# Gross changes in forest area shape the future carbon balance of tropical forests

Wei Li[1], Philippe Ciais[1], Chao Yue[1], Thomas Gasser[2], Shushi Peng[3], Ana Bastos[1]

[1]Laboratoire des Sciences du Climat et de l'Environnement, LSCE/IPSL, CEA-CNRS-UVSQ, Université Paris-Saclay, 91191 Gif-sur-Yvette, France
[2]International Institute for Applied Systems Analysis (IIASA), A-2361 Laxenburg, Austria
[3]Sino-French Institute for Earth System Science, College of Urban and Environmental Sciences, Peking University, Beijing 100871, China

*Correspondence to:* Wei Li (wei.li@lsce.ipsl.fr)

**Abstract.** Bookkeeping models are used to estimate land-use and land-cover change (LULCC) carbon fluxes ($E_{LULCC}$). The uncertainty of bookkeeping models partly arises from data used to define response curves (usually from local data) and their representativeness for application to large regions. Here, we compare biomass recovery curves derived from a recent synthesis of secondary forest plots in Latin America by Poorter et al. (2016) with the curves used previously in bookkeeping models from Houghton (1999) and Hansis et al. (2015). We find that the two latter models overestimate the long-term (100 years) vegetation carbon density of secondary forest by about 25%. We also use idealized LULCC scenarios combined with these three different response curves to demonstrate the importance of considering gross forest area changes instead of net forest area changes for estimating regional $E_{LULCC}$. In the illustrative case of a net gain in forest area composed of a large gross loss and a large gross gain occurring during a single year, the initial gross loss has an important legacy effect on $E_{LULCC}$ so that the system can be a net source of $CO_2$ to the atmosphere long after the initial forest area change. We show the existence of critical values of the ratio of gross area change over net area change ($\gamma_{Anet}^{Agross}$), above which cumulative $E_{LULCC}$ is a net $CO_2$ source rather than a sink for a given time horizon after the initial perturbation. These theoretical critical ratio values derived from simulations of a bookkeeping model are compared with real-world observations from the 30 m resolution Landsat TM data of gross and net forest area change in the Amazon. This allows us to diagnose areas where current forest gains with a large land turnover will still legate LULCC carbon emissions in 20, 50 and 100 years.

# 1 Introduction

The global carbon flux from land-use and land-cover change ($E_{LULCC}$) represents a net source of carbon to the atmosphere of $0.9 \pm 0.5$ Gt C $yr^{-1}$ during the last decade (Ciais et al., 2013; Le Quéré et al., 2015). $E_{LULCC}$ is usually estimated by bookkeeping models (Hansis et al., 2015; Houghton, 2003), dynamic global vegetation models (DGVMs) (Le Quéré et al., 2015; Sitch et
al., 2015) or compact earth system models (Gasser et al., 2017). Most DGVMs (e.g. in the TRENDY project, Sitch et al., 2015) estimate emissions due only to net area changes between different land-use / land-cover types in a grid cell. At the moment, efforts are being made to incorporate gross land-use and land-cover change (LULCC) in these models, that is for DGVMs the sub-grid transitions that sum up to net changes (Bayer et al., 2017). The bookkeeping model of Houghton (1999) includes emissions from both net area changes and gross LULCC from shifting cultivation, previously at the scale of large regions
(Houghton, 2003), and more recently for each country (Houghton and Nassikas, 2017). Gross LULCC occurs in tropical regions with shifting cultivation (Hurtt et al., 2011) and also in other regions where forests are cut and new plantations created at the same time. For example, consider a region with co-existing forest and crops where 20% of the land is converted from primary forest to crops while 20% sees crop abandonment to forest in the same period. The net change corresponds to a stable forest area, but the large carbon loss from primary forest is not compensated by the small carbon gain of the new plantations.
In this example, the region will be a net source of $CO_2$ during several years. Because of the non-symmetrical dynamics of $CO_2$ fluxes between forest loss and gain, $E_{LULCC}$ differs between net and gross area changes. Arneth et al. (2017) recently reviewed this issue using DGVMs and concluded that considering gross LULCC significantly increased the simulated $E_{LULCC}$ at global scale. Gross land-use area transition datasets including e.g. shifting cultivation practice (Hurtt et al., 2011) and reconstructions using empirical ratios between gross and net transitions (Fuchs et al., 2015) are now available and have been implemented in
a bookkeeping model (Hansis et al., 2015) as well as in some DGVMs to improve the estimate of $E_{LULCC}$ (Fuchs et al., 2016; Shevliakova et al., 2009; Stocker et al., 2014; Wilkenskjeld et al., 2014; Yue et al., 2017). However, uncertainties in the simulated $E_{LULCC}$ by grid-based DGVMs arise from the translation of the original LULCC datasets into plant functional type (PFT) maps and different processes comprised in different models (Arneth et al., 2017; Li et al., 2017). Although DGVMs are spatially and temporally explicit and include detailed physiological processes, the simulations using these models are time
consuming and require long spin-up simulations, small time step calculations of biophysical effects and carbon fluxes, including processes less relevant to $E_{LULCC}$. Thus, DGVMs are not appropriate to perform, for instance, sensitivity tests for the assessment of LULCC carbon fluxes.
Bookkeeping models use response curves for biomass and soil carbon stocks consecutive to LULCC disturbance and time-series of LULCC areas to estimate $E_{LULCC}$ (Hansis et al., 2015; Houghton, 1999). Response curves can be linear (Houghton,
1999, 2003), exponential (Hansis et al., 2015) or of other types. The carbon densities of different land-use types are derived from field measurements (Houghton et al., 1983). Even though carbon densities have a high spatial variability in the real world, the same response curve measured at one location is often applied in bookkeeping models over large regions. A recent study of the biomass resilience of secondary forests in the Neotropics provides new biomass recovery curves from 45 secondary

forest sites (Poorter et al., 2016). These new data are valuable to revisit the response curves for the regrowth of secondary forest in the Amazon area, an important region with a large $E_{LULCC}$.

In this study, we first aim to compare the recent biomass regrowth curves from Poorter et al. (2016) with the ones used in two bookkeeping models (Hansis et al., 2015; Houghton, 1999) for their implications in $E_{LULCC}$. Second, we demonstrate that

because of the asymmetry between carbon loss from deforestation and carbon gains from regrowth, even when the net forest area change is positive, a large initial gross forest area change can still cause $E_{LULCC}$ to be a source of $CO_2$ to the atmosphere on multi-decadal horizons. Last, we apply our conceptual calculation to the satellite forest data to diagnose areas with net forest gains but cumulative LULCC carbon emissions.

Based on $E_{LULCC}$ calculated using a bookkeeping approach and several idealized scenarios constructed to have different gross

forest area changes but with the same net area change (Section 3.2), we show the existence of a critical ratio of gross-to-net forest area change above which cumulative $E_{LULCC}$ remains a net source after initial LULCC, because carbon losses from deforestation are not compensated by carbon gains from secondary forest growth (Section 3.3). The theoretical value of this ratio derived from the idealized scenarios is then compared with actual estimates of gross-to-net forest area change over the Amazon derived from high-resolution (30 m) Landsat satellite imagery over the period of 2000-2012 (Hansen et al., 2013).

This allows us to identify sensitive regions where the current turnover of forest is too large, and may result in an emission source of $CO_2$ to the atmosphere over different time horizons in the future.

## 2 Methods

The land-use and land-cover changes considered in this study are forest loss (tropical moist forest transformed to cropland) and forest gain (cropland abandonment to secondary tropical moist forest) in Latin America. We construct a bookkeeping

model to simulate the carbon balance of simultaneous forest loss and gain in the same region. This model is similar to those developed by Houghton (1999) and Hansis et al. (2015) for global applications. After forest area loss, carbon density changes are calculated for biomass, two soil organic carbon pools (rapid and slow) and two products pools with turnover times of 1 and 10 years respectively. After the establishment of a secondary forest, carbon density changes in biomass and soil pools are considered. Only one slow soil pool is used in the regrowth of secondary forest, similar to Houghton (1999) and Hansis et al.

25  (2015).

Both the linear response curves from Houghton (1999) and the exponential ones from Hansis et al. (2015) are used to simulate the dynamics of each carbon pool consecutive to initial LULCC (Figure 1). For re-growing secondary forest, we also used two curves for biomass recovery based on a collection of field measurements by Poorter et al. (2016). The first one is a logarithmic equation describing aboveground biomass carbon as a function of stand age from Poorter et al. (2016), the parameters of which

are derived using the average aboveground biomass recovery from multiple stands after 20 years. It should be noted that with a logarithmic curve, no asymptotic value is reached even after an infinite time, which is not realistic for estimating long-term budgets, as it would mean permanent carbon gains. To overcome this problem of the logarithmic curve, we define a fixed time

horizon of 100 years after LULCC at which biomass becomes constant. The second biomass carbon gain curve is an exponential curve obtained by fitting the data from Poorter et al. (2016) with a saturating exponential function like in Hansis et al. (2015). This equation avoids the infinite increase of biomass after LULCC in the logarithmic curve. For both response curves, a ratio of 0.81 (Liu et al., 2015; Peacock et al., 2007; Saatchi et al., 2011) is used to convert aboveground biomass

reported by Poorter et al. (2016) to total biomass, and this ratio is consistent with the one (0.82) that Poorter et al. (2016) used based on FAO FRA report (FAO, 2010).

To model the sensitivity of the carbon balance of a typical region in Latin America to different ratios of gross-to-net forest area change during initial pulse of forest area change followed by no-change in forest area, we construct five idealized scenarios (Table 1). These scenarios are: S0 with no net but gross area changes; S1 with a net forest area loss being the sum of small

gross area changes; S2 with the same net forest area loss as S1, but being a sum of large gross area changes; and S3 and S4, similar to S1 and S2 but with a net forest area gain, instead of a net loss. An example of small versus large gross area changes with the same net area change is illustrated in Figure 2.

In each scenario, LULCC is applied as a pulse of forest area change at time $t = 0$, and we evaluate carbon changes over the following 100 years. The parameter $\gamma_{Anet}^{Agross}$ is the ratio of gross forest change area ($A_{gross}$) to net forest change area ($A_{net}$) applied

at $t = 0$.

$$\gamma_{Anet}^{Agross} = \frac{A_{gross}}{A_{net}} \tag{1}$$

where:

$$A_{gross} = |A_{loss}| + A_{gain} \tag{2}$$

$$A_{net} = A_{loss} + A_{gain} \tag{3}$$

By convention, $A_{loss}$ (<0) and $A_{gain}$ (>0) are the gross forest loss and gain areas applied at $t = 0$. A positive value of $A_{net}$ is an increase in forest area. For instance, the illustrative scenario S3 described in Table 1 explores the effects of a large positive value of $\gamma_{Anet}^{Agross}$ on $E_{LULCC}$. $E_{LULCC}$ is then simulated for contrasting $A_{gross}$ and $A_{net}$ transitions with the bookkeeping model as the sum of changes in all carbon pools over the area that was disturbed at $t = 0$. $\Sigma E_{LULCC,net}$ is the cumulative LULCC carbon flux up to a time horizon t, calculated using only net area changes ($A_{net}$) and ignoring gross area changes. $\Sigma E_{LULCC,gross}$ is the

cumulative carbon flux using gross forest area change, which has two component fluxes: the cumulative emissions ($\Sigma E_{LULCC,loss}$) from gross forest loss and the carbon sink ($\Sigma E_{LULCC,gain}$) from secondary forest regrowth. This is given by:

$$\Sigma E_{LUC,gross} = \Sigma E_{LUC,loss} + \Sigma E_{LUC,gain} \tag{4}$$

$$\Sigma E_{LUC,loss} = -A_{loss} \times L(t) \tag{5}$$

$$\Sigma E_{LUC,gain} = A_{gain} \times G(t) \tag{6}$$

where $L(t)$ and $G(t)$ stand for the cumulative carbon density change in all carbon pools up to time t. Positive values of carbon fluxes indicate a loss of land carbon to the atmosphere.

For each scenario in Table 1, we test different loss and gain response curves in our bookkeeping model, namely, linear or exponential carbon loss and linear, logarithmic or exponential increase for forest gain. In the case of gross forest area loss, we

considered two options, either a primary forest (primary-to-secondary) or a secondary forest (secondary-to-secondary) being cleared (Table 2, also see an illustration in Figure 2). This gives a total of eight combinations (C1 to C8 in Table 2) to calculate legacy $E_{LULCC}$ after a forest area disturbance. Note that one basic principle of bookkeeping models is that the same equilibrium vegetation carbon density is assumed between a secondary forest being lost and a secondary forest having fully recovered.

Therefore, the equilibrium biomass density of secondary forest being lost at t=0 in C1, C3 and C5 is set to be the same as that of the fully recovered (100 years) secondary forest in Poorter et al. (2016).

We use Global Forest Change data from Hansen et al. (2013) to apply our conceptual calculation to the real-world gross and net forest changes. Forest cover data from Hansen et al. (2013) comprise three layers at 30 m resolution: tree cover fraction (0-100% in each pixel) in year 2000, forest area loss (each pixel labeled with a loss year) during 2000-2012, and forest gain

during 2000-2012 (not specifying the gain year). As noted in Hansen et al. (2013), attributing the forest gain to a specific year is challenging because of the difficulty in detecting young forests from satellite reflectance measurements. In this study, we use the forest loss and forest gain layers to calculate the ratios of gross-to-net area changes ($\gamma_{Anet}^{Agross}$) at a 0.5° × 0.5° resolution, and thus $\gamma_{Anet}^{Agross}$ represents the average values during 2000-2012 rather than for a single year since the year of forest gain is not reported. The gross changes at the 0.5° level are calculated by summing the absolute areas of forest loss and gain at the 30 m

level during 2000-2012 in each 0.5° × 0.5° grid cell, while the net changes are the sum of gross forest loss (negative) and gross forest gain (positive).

## 3 Results

### 3.1 Response curves and comparison with field measurements

The response curves of tropical moist forest from bookkeeping models of Houghton (1999) and Hansis et al. (2015) and from

Poorter et al. (2016) for Latin America used in this study (Section 2) are displayed in Figure 1. The curves of Houghton (1999) (linear) and Hansis et al. (2015) (exponential) are similar (Figure 1) because the parameters of the exponential function were calibrated from the linear one (Hansis et al., 2015). Due to the higher carbon density of primary compared to secondary forest and the identical time at which both loss curves reach zero in Houghton (1999) and Hansis et al. (2015), the loss curves for a cleared primary forest are steeper than those for a cleared secondary forest (Figure 1a, b). This implies that clearing a primary

forest instead of a secondary one leads to larger legacy emissions. The fast decay of the rapid soil carbon pool in Figure 1a and 1b is due to the fact that a fraction of the initial biomass is assigned to this pool after forest clearing (Hansis et al., 2015; Houghton, 1999).

The logarithmic recovery curve (lime dashed lines in Figure 1c) from Poorter et al. (2016) has an initial faster biomass growth rate up to 20 years than in the curves used in previous bookkeeping models. After 20 or 30 years, however, the recovery curves

of Houghton (1999) and Hansis et al. (2015) surpass the one of Poorter et al. (2016), leading to a higher equilibrium biomass of mature secondary forests (Figure 1c). More precisely, the 100-year biomass of a secondary forest in Houghton (1999) and Hansis et al. (2015) is ≈ 25% higher than in Poorter et al. (2016). The median time to recover 90% of the maximum biomass

is 66 years in Poorter et al. (2016), compared to only 44 years in Houghton (1999) and 55 years in Hansis et al. (2015) (Figure 1c). The exponential recovery curve fit to the data from Poorter et al. (2016) (lime dash-dotted line in Figure 1c) has lower biomass than the logarithmic curve in the first 40 years but reaches a similar density after 100 years (by construction). The exponential curve from Poorter et al. (2016) agrees well with the linear curve of Houghton (1999) during the first 20 years
(Figure 1c).

## 3.2 Temporal change of cumulative carbon fluxes in different LULCC scenarios

We calculated cumulative carbon fluxes for the five idealized forest area change scenarios (Table 1) with the eight combinations of response curves (Table 2), giving an ensemble of 40 simulations. Results for each simulation are shown in Figure S1. We compare here the response curve combination C1 (exponential secondary forest loss and logarithmic biomass
recovery) and C2 (exponential primary forest loss and logarithmic biomass recovery) as examples in Figure 3 (see annual fluxes in Figure S2) to illustrate the effect of different gross forest area change with the same net area change on cumulative carbon flux, i.e., the impact of $\gamma_{A_{net}}^{A_{gross}}$ on the $E_{LULCC}$. Other combinations provide similar conclusions as C1 and C2. For example, $E_{LULCC}$ for C5 and C6 using linear curves for forest loss are very similar to C1 and C2 in Figure S1.

In the scenario S0 with initial secondary forests and no net forest area change, $\Sigma E_{LULCC,net}$ is zero when calculated based on net
area change (Figure 3a) but the gross carbon flux ($\Sigma E_{LULCC,gross}$) is distinct from zero. In the variant of the S0 scenario with initial primary forest (C2), due to the lower equilibrium carbon density of the secondary forest, $\Sigma E_{LULCC,gross}$ is a large source after 100 years (red dashed lines in Figure 3a). In the secondary forest loss and gain case (C1), $\Sigma E_{LULCC,gross}$ is a carbon source in the initial period and gradually becomes carbon neutral with the compensation effects of secondary forest regrowth (red solid lines in Figure 3a).

Both S1 and S2 scenarios have the same net forest area loss ($A_{net}$ = -1 ha) but different gross forest area changes ($\gamma_{-1}^{1,2}$ = -1.2 and $\gamma_{-1}^{201}$ = -201 for S1 and S2 respectively, Table 1). In S1 with a small gross area change ($A_{gross}$ = 1.2 ha), $\Sigma E_{LULCC,gross}$ is close to $\Sigma E_{LULCC,net}$ (Figure 3b), starting with either primary and secondary initial forests. By contrast, the difference between $\Sigma E_{LULCC,gross}$ and $\Sigma E_{LULCC,net}$ in S2 is large and positive, indicating a cumulative carbon loss much higher than S1 due to its large gross area change (Figure 3c).

The scenarios S3 versus S4 with a net forest gain ($A_{net}$ = +1 ha) but different ratios of gross-to-net area changes ($\gamma_{A_{net}}^{A_{gross}}$) present a similar behavior as S1 versus S2. However, the sign of $\Sigma E_{LULCC,gross}$ is reversed, from a sink in S3 (red lines in Figure 3d) to a source in S4 (red lines in Figure 3e). Especially for the gross primary forest loss, $\Sigma E_{LULCC,gross}$ exhibits a large source even after 100 years (red dashed lines in Figure 3d,e). This implies that despite the net initial forest gain, the rate of gross area change determines the sign of $\Sigma E_{LULCC}$ over a certain time horizon after the pulse of forest area change. More generally, this
shows that, while long term cumulative land use change emissions are determined only by the net land use area change (e.g. Gasser and Ciais, 2013), short term cumulative emissions are determined by the gross area change.

### 3.3 Change of $\Sigma E_{LULCC,gross}$ with the same net forest gain but different gross area changes

The comparison of $\Sigma E_{LULCC,gross}$ (Figure 3) for the idealized scenarios (Table 1) illustrates the fact that different values of $\gamma_{Anet}^{Agross}$ have a large impact on the magnitude and the sign of cumulative LULCC emissions depending on the time elapsed after the initial pulse of forest area change. We thus calculated the difference between $\Sigma E_{LULCC,gross}$ and $\Sigma E_{LULCC,net}$ by varying $\gamma_{Anet}^{Agross}$ in

a systematic manner in a net forest gain scenario (Figure 4).

When $\gamma_{Anet}^{Agross}$ is increased, i.e., with more forest land turnover at t = 0 for the same initial net forest area gain ($A_{net}$ = +1 ha), the time for $\Sigma E_{LULCC,gross}$ to become a net carbon sink becomes longer (Figure 4a). With initial primary forest being cut at t = 0, the cumulative LULCC carbon flux is still a source of $CO_2$ to the atmosphere after 100 years, even in simulations where the net forest area was increased at t = 0 (Figure 4b). This highlights that the different initial carbon density of primary forest from

secondary forest can lead to very long-term legacy emissions.

The critical value of $\gamma_{Anet}^{Agross}$ that reverses the sign of $\Sigma E_{LULCC,gross}$ from carbon source to sink increases as a function of the time-horizon considered after the initial forest area change (Figure 4c). The two cases with initial secondary and primary forest loss show a different trajectory of this ratio along time. In the former, $\gamma_{Anet}^{Agross}$ increases slowly in the beginning and then sharply, while in the latter $\gamma_{Anet}^{Agross}$ increases quickly at the initial stage and then at a smaller rate. In fact, if $\Sigma E_{LULCC,gross}$ can reach zero

(the point of sign changed, let $\Sigma E_{LULCC,gross}$ = 0), combining with equations (1) to (6), the critical value of $\gamma_{Anet}^{Agross}$ can be expressed as:

$$\gamma_{Anet}^{Agross} = \frac{L(t)-G(t)}{L(t)+G(t)} \tag{7}$$

This critical value of $\gamma_{Anet}^{Agross}$ is independent of the initial forest area but determined by the carbon density changes at a given time consecutive to a change of forest area. Thus, for a secondary forest loss and gain at t = 0, the long-term L(t) + G(t) tends

to zero and $\gamma_{Anet}^{Agross}$ goes to infinite. For a primary forest loss and secondary forest gain at t = 0, the long-term L(t) + G(t) is the difference in the equilibrium carbon densities between primary and secondary forest, and therefore $\gamma_{Anet}^{Agross}$ approaches a constant value at t = infinite. Furthermore, it should be noted that our approach of analyzing the critical value of $\gamma_{Anet}^{Agross}$ is not limited to net forest gain scenarios or to LULCC transitions between forest and cropland. The framework of $\gamma_{Anet}^{Agross}$ can also be extended to other LULCC scenarios, including lower, higher, and equal equilibrium biomass density between two land-use types. For

example, if a re-growing forest can achieve a higher equilibrium carbon density than the initial one, there is also a critical $\gamma_{Anet}^{Agross}$ for the net forest loss scenario, for which the gross carbon emission becomes a sink at a certain time after initial forest area change. This situation may happen in reality, if the deforested forests are replaced by more productive species or under active management like fertilization and irrigation. Even in the field measurements by Poorter et al. (2016), some Neotropical secondary forests show very high biomass resilience, i.e., reaching to a higher biomass than pre-deforestation.

We also calculated the critical ratios over time based on the exponential biomass response curves from Hansis et al. (2015) in comparison with the response curves from Poorter et al. (2016) (Figure S3). As show in Figure 1, the equilibrium of secondary forest vegetation density with the recovery curve of Hansis et al. (2015) is higher than with Poorter et al. (2016) and we assumed that the same density of primary forest for both, and thus $L_{Hansis,primary}(t) = L_{Poorter,primary}(t)$, $L_{Hansis,secondary}(t) >$

$L_{Poorter,secondary}(t)$ and $G_{Hansis}(\infty) > G_{Poorter}(\infty)$. Note that a positive value of carbon flux indicates carbon emission to the atmosphere. Combined with Eq (7), the different equilibrium states of secondary forest vegetation can explain the differences of critical ratios over time between Hansis et al. (2015) and Poorter et al. (2016) in Figure S3.

## 3.4 Ratios in Latin America from satellite imagery

Based on the theoretical evidence for the existence of a critical value of the gross-to-net forest area change ratio ($\gamma_{Anet}^{Agross}$), which determines the sign and magnitude of $\Sigma E_{LULCC,gross}$ at a given time after an initial net forest area change, we further combined such ratios with the land-cover change dataset to determine whether a region is a carbon sink or source at a given time horizon. Using the 30 m resolution forest area change data of Hansen et al. (2013) between 2000 and 2012, we calculated the ratios ($\gamma_{Anet}^{Agross}$) at a spatial resolution of $0.5° \times 0.5°$ in the same region of Latin America as Poorter et al. (2016). The spatial resolution

of $0.5° \times 0.5°$ is a typical resolution of DGVMs when they simulate global $E_{LULCC}$. We set a future time horizon of 20 years as that is close to the targeted year in the Nationally Determined Contributions (NDCs) (Grassi et al., 2017). From Figure 4c, the critical values of $\gamma_{Anet}^{Agross}$ at 20 years after an initial change in forest area are 7.2 and 2.4 respectively for secondary-to-secondary and primary-to-secondary initial transitions. For a longer time horizon of 50 years, the critical values are 22.5 and 3.1, respectively. After 100 years of the initial forest area change, while the critical value of $\gamma_{Anet}^{Agross}$ for secondary-to-secondary

transition goes to infinite, it approaches a constant value of 3.7 for primary-to-secondary forest change (Figure 4c).

The map of $\gamma_{Anet}^{Agross}$ diagnosed from the 30 m Landsat forest cover data in grid cells of $0.5° \times 0.5°$ is shown in Figure 5. Note that here we focus only on the grid cells with a net forest gain. The number of $0.5° \times 0.5°$ grid cells where $\gamma_{Anet}^{Agross} > 7.2$, that is grid cells where current forest area change will lead to a source of $CO_2$ over a 20-year horizon, is 102 in our domain (Figure 5a), which accounts for 35% of the total number of grid cells where a net forest gain is observed between 2000 and 2012. In these

102 grid cells, the $\Sigma E_{LULCC,gross}$ is simulated to be a cumulative carbon emission in 20 years, no matter whether the lost forest is primary or secondary. If primary forests are cleared in grid cells with $2.4 < \gamma_{Anet}^{Agross} < 7.2$ (33% of the total forest gain grid cells, Figure 5a), the 20-year $\Sigma E_{LULCC,gross}$ is also a carbon source rather than a sink. We note that it is not possible to separate the primary and secondary forest in the forest cover data of Hansen et al. (2013), so we cannot say whether these grid cells with $2.4 < \gamma_{Anet}^{Agross} < 7.2$ are carbon source or sink in the real world. For a time horizon of 50 years, the fractions of grid cells

with $\gamma_{Anet}^{Agross} > 22.5$ and with $3.1 < \gamma_{Anet}^{Agross} < 22.5$ in total net forest gain grid cells are 14% and 46% respectively (Figure 5c). The 100-year $\Sigma E_{LULCC,gross}$ in grid cells with $\gamma_{Anet}^{Agross} > 3.7$ (53% of total) is also possible to be a carbon source if lost forest is primary in these grid cells (Figure 5d). The grid cells with $\gamma_{Anet}^{Agross}$ greater than the critical values are mainly distributed in Southeast Brazil (Figure 5b,c,d).

By comparison, we also calculated the number of grid cells with $\gamma_{Anet}^{Agross}$ above the critical ratio for the biomass response curves

from Hansis et al. (2015) (Table S1). Because of the differences in the critical values of $\gamma_{Anet}^{Agross}$ over time (Figure S3) between curves from Poorter et al. (2016) and Hansis et al. (2015), a higher critical ratio leads to smaller number of $0.5° \times 0.5°$ grid cells with $\gamma_{Anet}^{Agross}$ beyond the critical ratio (Table S1).

In addition to the number of grid cells with $\gamma_{Anet}^{Agross}$ above the critical ratio, we further showed the differences between the cumulative carbon flux using gross transitions ($\Sigma E_{LULCC,gross}$) and net transitions ($\Sigma_{ELULCC,net}$) in these grid cells (Table S2). Taking C1 (secondary-to-secondary) at 20 yr horizon for example, using net transitions results in a carbon sink of 12 Tg C but using gross transitions results in a carbon emission of 21 Tg C (Table S2) in the grid cells with $\gamma_{Anet}^{Agross} > 7.2$ (Figure 5b).

**4 Discussion**

The biomass recovery curves of Neotropical secondary forests from Poorter et al. (2016) are lower 20 years after the initial perturbation than those used in the bookkeeping models of Houghton (1999) and Hansis et al. (2015), implying that these models simulate different LULCC carbon fluxes in Latin America from those using the recovery curves of Poorter et al. (2016). The carbon density in undisturbed forests in the bookkeeping models of Houghton (1999) and Hansis et al. (2015) were

essentially based on Whittaker and Likens (1973), multiplied by a factor of 0.75 to approximate the lower carbon density of secondary forests (Houghton et al., 1983). The carbon density data from Whittaker and Likens (1973) are subject to two sources of uncertainty. First, these values represent biomass in the 1950s (Woodwell et al., 1978) rather than present days, and second, they were compiled from very limited field measurements for tropical forests. In fact, Whittaker and Likens (1973) claimed in their study that data "for tropical communities are very meager" and the mean biomass density is "a subjectively chosen

intermediate value based on very few measurements" to avoid extreme values.

Differences may also exist for soil carbon dynamics after LULCC. There are a great number of meta-analyses or reviews (Conant et al., 2001; Davidson and Ackerman, 1993; Davis and Condron, 2002; Don et al., 2011; Guo and Gifford, 2002; Kurganova et al., 2014; Laganière et al., 2010; Li et al., 2012; Marín-Spiotta and Sharma, 2013; Murty et al., 2002; Paul et al., 2002; Poeplau et al., 2011; Post and Kwon, 2000; Powers et al., 2011; Wei et al., 2014; West et al., 2004) on the soil carbon

change after LULCC based on field measurement data (mostly paired sites and chronosequences). These studies may generally agree with the directions of soil carbon change after LULCC (e.g. soil carbon loss after forest clearing for cropland), but the magnitudes and temporal dynamics of soil carbon changes remain highly uncertain because, among other things, of the limited site number and the diversity of soil properties. Field measurements at site level may be unrepresentative of the whole region because the distribution of biophysical conditions like soil texture, precipitation and temperature may not match the distribution

of the whole set of such factors in the LULCC areas in a given region (Powers et al., 2011).

Some DGVMs (Bayer et al., 2017; Shevliakova et al., 2009; Stocker et al., 2014; Wilkenskjeld et al., 2014; Yue et al., 2017) as well as a bookkeeping model Hansis et al. (2015) have implemented gross land use and land cover transitions, and thus simulated a higher $E_{LULCC}$ than using net transitions. Arneth et al. (2017) reviewed the "missing processes" in LULCC modeling by DGVMs and found that ignoring gross LULCC could underestimate the global $\Sigma E_{LULCC}$ by 36 Pg C on average

over the historical period (1901-2014). In this study, we used a bookkeeping method to quantify the difference in LULCC emissions calculated using net versus gross forest area transitions, and to show the existence of critical ratios of gross-to-net forest area changes above which land use action will cause a reversed sign of cumulative carbon flux. Evidently, the choice of

a time horizon to assess the carbon balance of a system after an initial pulse of forest area change influences the value of the critical ratio $\gamma_{\text{Anet}}^{\text{Agross}}$. The desirable target time lengths could be different depending on specific mitigation projects or land-use reduction policies, and thus critical values of the gross-to-net forest area change ratio are different (Figure 4c). On the other hand, because of the temporal evolution of legacy carbon fluxes after initial land disturbance, it is important to define a specific
and reasonable time horizon when making land-based mitigation policies.

As a conceptual analysis, the assumptions we made raise uncertainties. First, the logarithmic biomass recovery curve adopted in Poorter et al. (2016) does not seem to be appropriate for LULCC emission modelling because it does not reach an equilibrium state. We thus fitted the data from Poorter et al. (2016) with an exponential saturating curve to avoid this issue. Second, we used a median biomass recovery rate for the whole tropical moist forest region in Latin America. In reality, however, due to
the different climate, soils and other ecosystem conditions, recovery rates vary, and thus spatially explicit recovery rates should better depict regional patterns of secondary forest regrowth and net LULCC emissions. In the dry tropics, the critical ratio values may be smaller because of the slower biomass recovery rates. Third, the biomass and soil carbon densities in initial vegetation and the equilibrium vegetation after LULCC are also spatially different in the real world. The distinction between primary and secondary forest being lost at $t = 0$ is a typical example of how different initial carbon density impacts the legacy
LULCC carbon flux and thus the determined critical gross-to-net ratio values. In fact, a large spatial gradient of biomass exists from Northeast to Southwest Amazon region (Saatchi et al., 2007, 2011). One possible approach to account for the spatial variations of both biomass recovery rate and biomass density would be to reconstruct spatially explicit biomass–age curves using relationship between regrowth rates and climate (Poorter et al., 2016) and to combine with observation-based biomass densities (Baccini et al., 2012; Saatchi et al., 2011) and satellite-based forest cover change (Hansen et al., 2013). However,
uncertainties arise in the up-scaling of biomass recovery rates and lack of information on annually resolved forest gain from Hansen et al. (2013). In addition, spatially explicit soil carbon density maps are also uncertain.

The effect of gross-versus-net forest area change on legacy LULCC emissions certainly differs across forest ecosystems and other LULCC transition types (e.g. transitions between grassland and cropland). The concept of critical ratios of gross-to-net LULCC affecting legacy carbon balance can be extended in other regions where forest management practice is critical (e.g.
North America and Europe). Forest management practices like wood harvest and thinning extract carbon from the ecosystem and release it to the atmosphere (Houghton et al., 2012), while recovering secondary forest from past deforestation and logging (Pan et al., 2011) and even old-growth forests (Luyssaert et al., 2012) can act as carbon sinks. In theory, likewise, a critical ratio value should exist to balance the bi-directional carbon fluxes in forest management practices. An advantage of this concept of critical ratio is that it can be directly measured with satellite observations, which provides a quick guide for local land use
management practice through near-real-time forest cover change data (e.g. Global Forest Watch http://www.globalforestwatch.org/).

Accurate estimates of LULCC carbon fluxes in the Neotropical forests are increasingly important for climate mitigation policy with the progressive implementation of Reducing Emissions from Deforestation and forest Degradation (REDD+) programs under the United Nations Framework Convention on Climate Change (UNCCC) (Angelsen et al., 2009; Magnago et al., 2015).

Furthermore, forest-based climate mitigation has been taken as a key option in the Nationally Determined Contributions (NDCs) proposed by some countries to the Paris Climate Agreement, accounting for about one-fourth of total intended emission reductions from a pre-defined baseline (Grassi et al., 2017). Brazil contributes about one-third of the global forest-based emission reduction in the NDCs (Grassi et al., 2017). Based on the results of this study, we argue that it will be important

to carefully distinguish the amount of gross vs. net forest changes and clearing of primary vs. secondary forest when assessing national forest-based mitigation pledges. With a large gross to net area change ratio, a net forest gain could still legate a net carbon source over a long period in the future. Our work has the potential to be extended to country-level and other LULCC types as long as information on vegetation and soil carbon densities changes after LULCC is available, and a critical value of $\gamma_{Anet}^{Agross}$ can be estimated as a guideline to evaluate land-based mitigation policies for each region. More observation-based data

on land-use area change and carbon loss and gain curves will definitely help to extend the range of applications of the critical gross-to-net area ratio concept.

**5 Conclusions**

Using only net LULCC transitions instead of gross values can bias the magnitude of estimated LULCC carbon fluxes, to the point of estimating a sink instead of a source in reality if high gross forest area change occurs. We used idealized scenarios to

demonstrate different aspects of the discrepancy between net and gross forest changes, defining the $\gamma_{Anet}^{Agross}$ metric as the ratio of gross area change to net area change. Our S0 experiment shows even that there is no net forest change, LULCC may actually lead to a carbon source, depending on the gross forest change area. S1 and S2 show that with the same net forest loss, different ratios of gross-to-net forest change ($\gamma_{Anet}^{Agross}$) alter the magnitude of differences between net and gross cumulative carbon fluxes. Similarly, S3 and S4 show that with the same amount of net forest gain area, different $\gamma_{Anet}^{Agross}$ can even change the directions of

carbon fluxes, i.e. from a gross carbon sink to source even that net forest area increases. We further determined the critical ratios in net forest gain grid cells ($\gamma_{Anet}^{Agross} = 7.2$ and 2.4 respectively for secondary and primary forest clearing), above which the gross cumulative carbon fluxes show a reversed sign than the net ones at 20 years after LULCC occurred. These analyses reveal the importance of using gross LULCC transitions rather than net LULCC transitions in both bookkeeping models and DGVMs. The concept of critical ratio can be also implemented in other LULCC transitions in other regions and used as a

guide for carbon balance estimation in forest management.

**Acknowledgements**

W.L., P.C., T.G. and S.P. acknowledge support from the European Research Council through Synergy grant ERC-2013-SyG-610028 "IMBALANCE-P". W.L. and C.Y. are supported by the European Commission-funded project LUC4C (No. 603542).

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

**Table 1** Illustrative scenarios with different ratios of gross-to-net forest area changes impacting legacy LULCC emissions after a pulse disturbance of forest area at t = 0. $A_{net}$, $A_{gross}$, $A_{loss}$, $A_{gain}$ and $\gamma_{A_{net}}^{A_{gross}}$ are the applied net forest area change, gross forest area change, gross forest loss area, gross forest gain area and the ratio of $A_{gross}$ to $A_{net}$ at t = 0. Positive value of an area change is an increase of forest area.

| Scenario | $\gamma_{A_{net}}^{A_{gross}}$ | $A_{net}$ (ha) | $A_{gross}$ (ha) | $A_{loss}$ (ha) | $A_{gain}$ (ha) |
|---|---|---|---|---|---|
| S0 | $\gamma_0^2 = \infty$ | 0 | 2 | -1 | 1 |
| S1 | $\gamma_{-1}^{1.2} = -1.2$ | -1 | 1.2 | -1.1 | 0.1 |
| S2 | $\gamma_{-1}^{201} = -201$ | -1 | 201 | -101 | 100 |
| S3 | $\gamma_1^{1.2} = 1.2$ | 1 | 1.2 | -0.1 | 1.1 |
| S4 | $\gamma_1^{201} = 201$ | 1 | 201 | -100 | 101 |

**Table 2** Different combinations of response curves to calculate $E_{LULCC}$.

| Combination | Forest loss forest type | response curve in all carbon pools | Forest gain forest type | response curve for biomass | response curve for soil |
|---|---|---|---|---|---|
| C1 | secondary | exponential, Hansis | secondary | logarithmic, Poorter | exponential, Hansis |
| C2 | primary | exponential, Hansis | secondary | logarithmic, Poorter | exponential, Hansis |
| C3 | secondary | exponential, Hansis | secondary | exponential, Poorter | exponential, Hansis |
| C4 | primary | exponential, Hansis | secondary | exponential, Poorter | exponential, Hansis |
| C5 | secondary | linear, Houghton | secondary | logarithmic, Poorter | exponential, Hansis |
| C6 | primary | linear, Houghton | secondary | logarithmic, Poorter | exponential, Hansis |
| C7 | secondary | exponential, Hansis | secondary | exponential, Hansis | exponential, Hansis |
| C8 | primary | exponential, Hansis | secondary | exponential, Hansis | exponential, Hansis |

**Figure 1** Response curves for tropical moist forest in bookkeeping models and from a recent field study. Solid and dotted lines indicate the linear (Houghton, 1999) and exponential (Hansis et al., 2015) curves, respectively. Lime dashed and dash-dotted lines are the logarithmic and exponential curves from forest plots (Poorter et al., 2016). Vegetation carbon density in primary forest (Houghton, 1999) is also shown as a star in (c) for comparison.

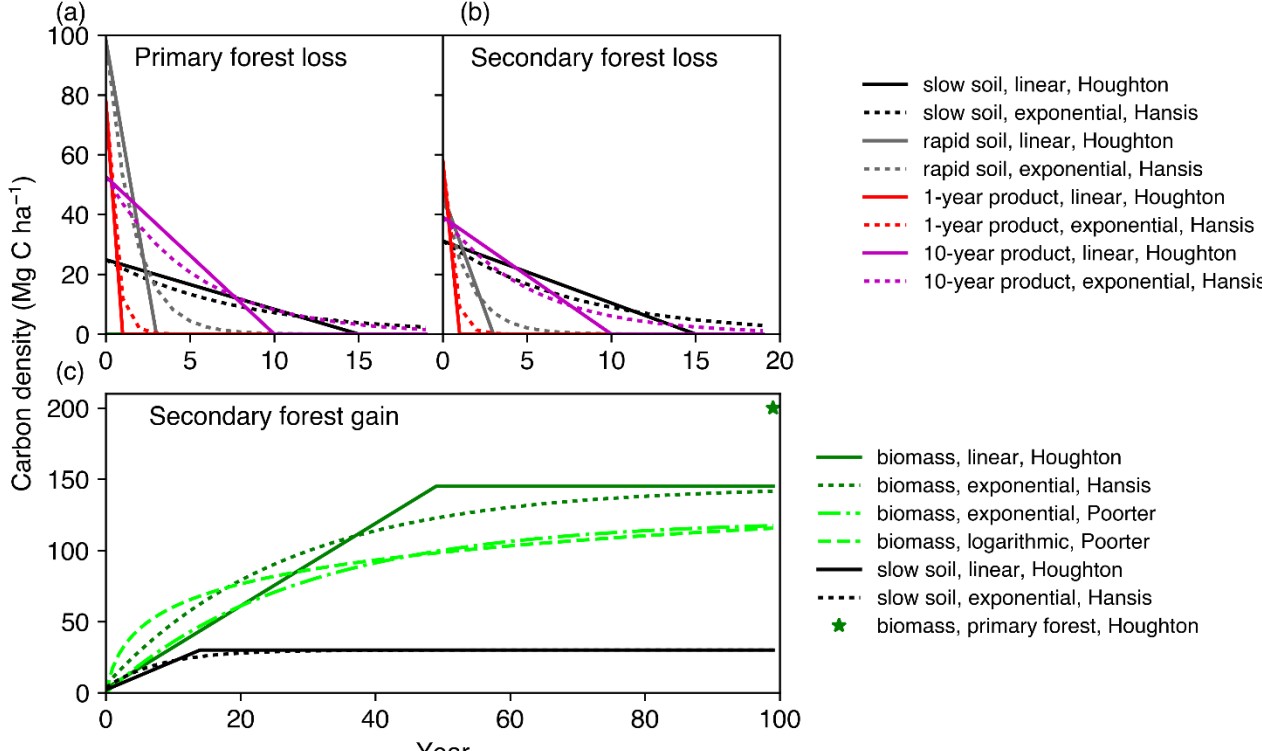

**Figure 2** An illustration of different gross forest area changes with the same net area change. (a) Net forest gain with small gross secondary forest area changes (secondary-to-secondary), thus low $\gamma_{Anet}^{Agross}$. (b) Same net forest gain as (a) but with large gross secondary forest area changes (secondary-to-secondary), thus high $\gamma_{Anet}^{Agross}$. (c) Same as (a) but with gross primary forest loss (primary-to-secondary) instead of gross secondary loss.

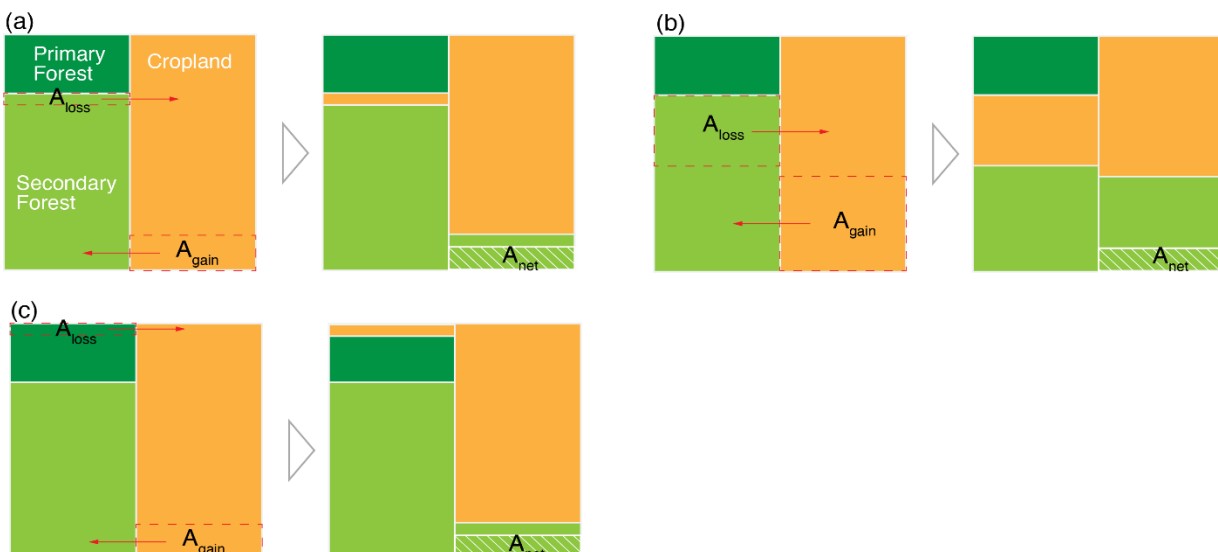

**Figure 3** Cumulative carbon flux ($\Sigma E_{LULCC}$) after an initial forest area change at $t = 0$ followed by no change in forest area, for the different scenarios S0 to S4 in Table 1 with different net and gross initial forest area changes. The response curves used in those bookkeeping model simulations are C1 in solid lines (Table 2) with a secondary-to-secondary forest change at $t = 0$ and a logarithmic biomass recovery curve with an asymptote, and C2 in the dashed lines (primary-to-secondary forest change at t = 0 and a logarithmic biomass recovery curve with an asymptote). The dotted line is the zero line. Positive value of carbon flux indicates carbon emission to the atmosphere.

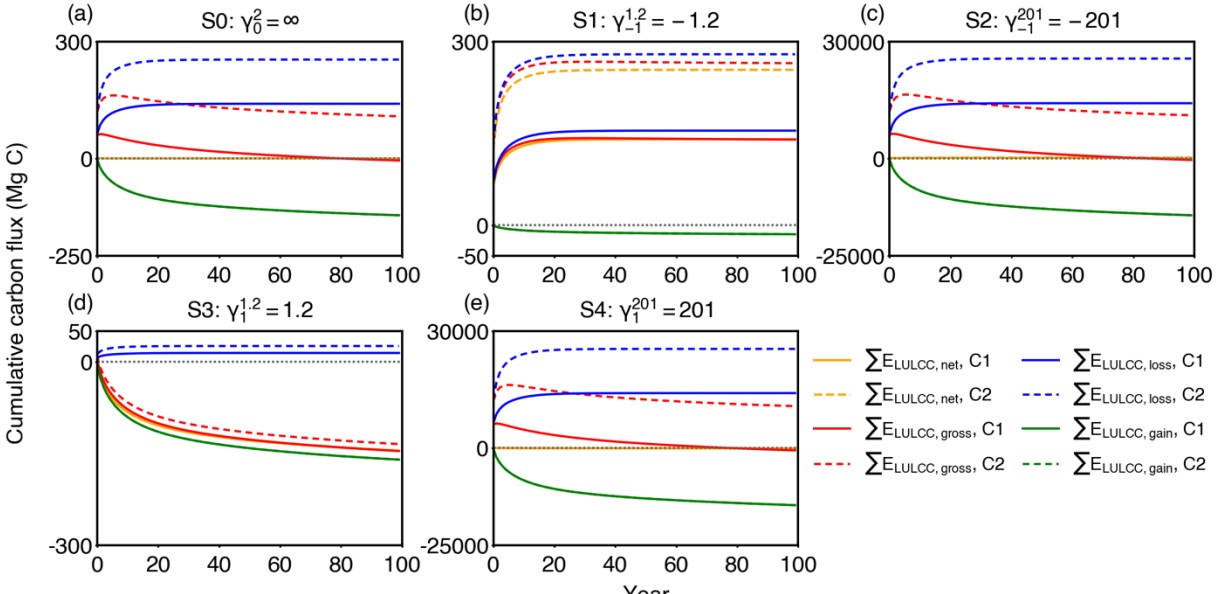

**Figure 4** Time evolution of cumulative carbon flux ($\Sigma E_{LULCC,gross}$) after an initial forest area change involving gross forest area changes followed by no forest area change. The three panels show results of our bookkeeping model for three case studies (a) a net forest gain at t = 0 with initial secondary forest loss followed by secondary forest regrowth (secondary-to-secondary, C1 in Table 2), (b) the same net area gain at t = 0 with initial primary forest loss followed by secondary forest regrowth (primary-to-secondary C2 in Table 2), and (c) the critical value of $\gamma_{Anet}^{Agross}$ at which $\Sigma E_{LULCC,gross}$ is zero, going from a net source to a net sink for different time horizon in the x-axis. The colored curves in (a) and (b) have the same net area change ($A_{net}$ = +1 ha) at t = 0 but variable values of the initial gross-to-net area change ratios ($\gamma_{Anet}^{Agross}$). The red line in (a) and (b) is the zero line, defining the time after initial forest area change at which the system reaches a neutral carbon balance. The light and dark green lines in (c) represent the critical ratios for a net initial forest gain scenario with secondary-to-secondary (a) and primary-to-secondary (b) gross forest area change, respectively. Values larger than this critical value indicate that the initial forest area change has the net effect to emit $CO_2$ for a given time horizon in the x-axis. Exponential curve from Hansis et al. (2015) for carbon loss in all pools and gain in soil pool and logarithmic curve from Poorter et al. (2016) for gain in biomass pool are used in this example (Table 2).

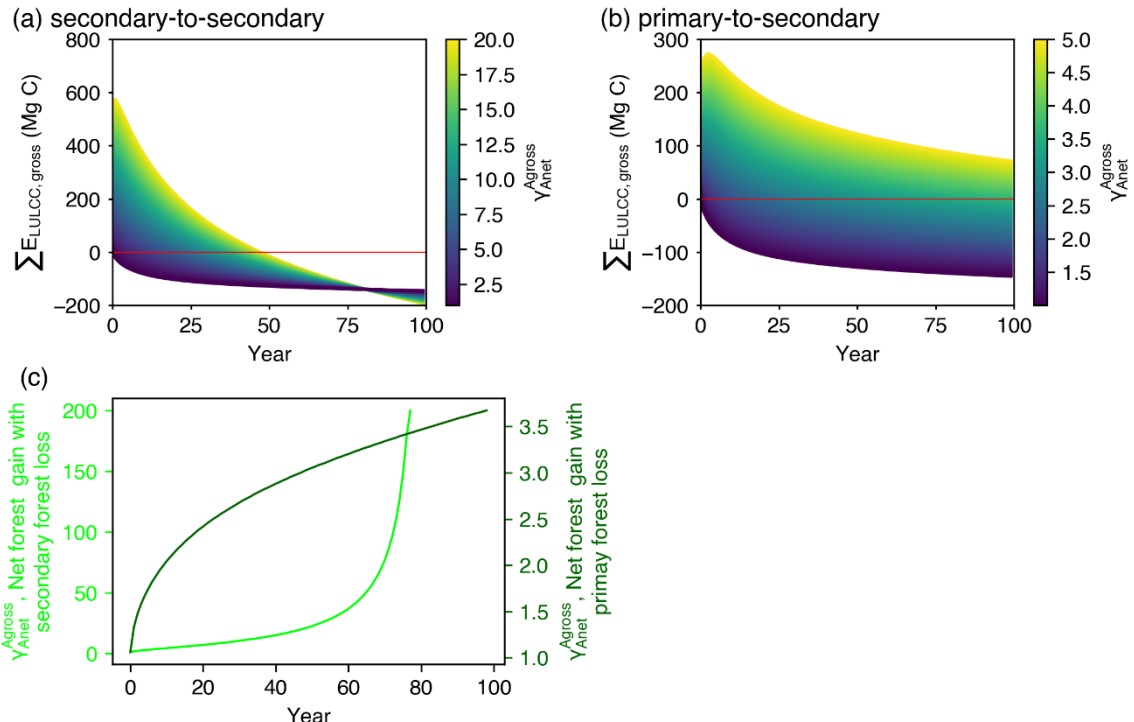

**Figure 5** Ratios of gross-to-net forest area change ($\gamma_{Anet}^{Agross}$) in 0.5° × 0.5° grid cells in Latin America (same region as Poorter et al. (2016)) calculated from the high-resolution forest cover change map (Hansen et al., 2013). Grid cells with $\gamma_{Anet}^{Agross} < 2.4$ are masked. (b) is the zoom-in area of 20-30° S and 40-60° W in (a) (red rectangle) and grid cells with $\gamma_{Anet}^{Agross} > 7.2$ and with $2.4 < \gamma_{Anet}^{Agross} < 7.2$ are shown as blue and green respectively to indicate those beyond the critical ratios with a time horizon of 20 years. (c) and (d) are similar to (b) but indicate a time horizon of 50 and 100 years respectively. The blue grid cells in (b) and (c) represent a cumulative carbon emission in 20 years no matter whether the lost forest is primary or secondary. The green ones in (b), (c) and (d) represent a cumulative carbon emission only if the cleared forests are primary forests.

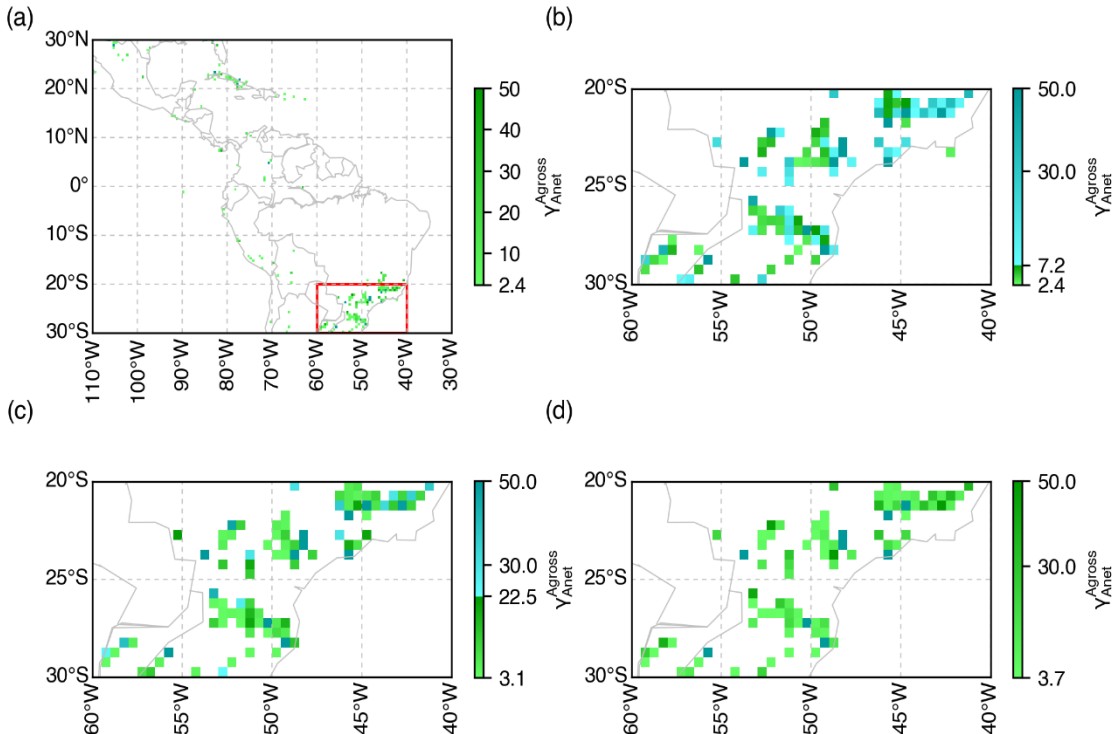