# Peer review of "Gross changes in forest area shape the future carbon balance of tropical forests"

_Biogeosciences, 2017_

## Referee Comment (RC1) · Anonymous Referee #1 · 26 Sep 2017

General comments

This is an interesting study, pointing out the importance of using gross instead of net land use transitions, distinguish between clearing of primary vs. secondary forest and to define a specific and reasonable time horizon when making land-based mitigation policies. Three main steps were taken: 1) the comparison between different response curves, 2) calculating different theoretical scenarios with a bookkeeping model to show the importance of considering gross forest area change and finding critical and 3) applying the ratio to real net land cover transitions from satellite data. Thereby, step 2 clearly takes the highest priority and consideration.

Still, some revision is needed: the abstract is very long which makes it partly difficult to get the main message of the study. Also, the gap in current research is not carved

out very well (page 2, ln 16 ff, says that other models have already implemented gross transitions) and the objectives should become clearer. In the introduction a two-fold purpose of the study is mentioned, what about the 3rd step? What was its objective? The 3rd point cannot be found in the method section, it is just roughly described in the results. Thereby some steps remain unclear: e.g. the model considers LC transition to take place at time = t0, but the satellite covers a time series of 12 years. Are all the transitions during these years threatened as if they took place at one time t=0 and then the results for the different time horizons of 20, 50 and 100 years are calculated based on that? Or is the exact time of each transition considered and the time horizons starts to be calculated after the last transition took place? Or do the gross transitions in this case refer not to time (i.e. shift of one LC to another LC and back) but instead refer to transitions within the calculated gridcells of 0.5° resolution, as the satellite data was mentioned to have a 30 m resolution?

One possibility to handle the dominance of point 2 would be to make it to the only objective, and shift point 1 to the method section – the comparison seems to be anyhow just a plot of the different curves that justifies the usage of response curves based on Poorter instead of those from Houghton and Hansis. Another possibility would be to include 3 in the method section and give point 1 and 3 more weight - e.g. by calculating the critical gross to net ratios based on the Houghton and Hansis functions, applying it to the same grid cells and comparing it then with the results based on the Poorter function. This would be interesting outcome and extend the first objective of comparing the different response curves to more than just a simple plot of the different curves in the same graph. Further, it would be very interesting to not only know whether each gridcell was a sink or source but also to quantify the ELUC and sum it to total number – one if everything was primary forest at the first transition, and one as if all was secondary, and the same if the other response curves were used.

Specific comments

Abstract: The abstract should be shortened to better focus on the findings, which would

make it easier to read and understand. E.g. is the 3rd sentence really relevant for the findings of this study? Especially also from line 19 to 27 there might be possibilities to shorten, summarize and simplify. Where shapes of the three different curves relevant for the finding? The finding here is difficult to understand, the sentences a bit complicated and several sentences basically say the same: You found and show critical values of gross to net forest area change above which ELUC of a net a net forest area gain switches from CO2 sink to source.

Introduction: Page 2, ln 8: that is for DGVMs the sub-grid transitions that sum up to net changes, here a reference to e.g. Bayer et al. 2017: doi:10.5194/esd-8-91-2017 could be nice, who focused on the problematic of sub-grid transitions.

3.4 Page 7, ln 14: "we pose the question whether such ratios can be observed in the real world" – but this is not what you are answering with your approach. As far as I understood you just calculate using your rates, whether the regions are a sink or a source.

Page 7, ln 22: "With a too high rotation rate of forests, i.e. a large gross to net area change ratio, a net forest gain could still legate a net carbon source over a long period in the future." I don't agree, as I think long rotation secondary forests should have other response curves than short rotation forest, as short rotation forest don't store as much carbon that can be lost afterwards.

4. discussion You state that the response curves used in bookkeeping models from Houghton (1999) and Hansis et al. (2015) overestimate carbon density – that implies that Porters values are true, while Houghton and Hansis are wrong. But also Houghton and Hansis are based on measurements, right? Maybe just not in the right region? It would be helpful to mention in the discussion where the measurements for Houghton and Hansis models were located.

Is the extent to which gross versus net transitions affect ELUC comparable what other studies investigating gross versus net transitions studied? You mention several studies

about this issue which were performed with carbon models – they should also appear in the discussion, showing how your results compare with what they found.

Technical corrections

General: In all figures and the text, it might be useful to replace the term "biomass carbon density" by "vegetation carbon density", as biomass is less well defined and includes in some disciplines also dead biomass and soil microbial biomass which would count here to the soil pool.

Page 1, ln 26: "critical value" should be plural

Page 2, Ln 10: "Gross LUC occurs in tropical regions with shifting cultivation (Hurtt et al., 2011) but also everywhere forests are cut and new plantations created at the same time" Is here a "where" after "everywhere" missing?

Table 1: The table caption uses gamma gross to net but in the table heading and the text gamma Agross to Anet is used.

Fig 1: biomass from primary forest: reference missing; Legend for a) and b) shows biomass, which can only be found in plot c) that has an own legend.

Fig 3: include dashed and solid line in legend. In figure description logarithmic asympthotic should be removed or refered to both – solid and dashed, so it becomes more clear that there is no difference in the response curve between solid and dashed, but just which systems are transformed.

Fig 4 is a bit difficult to understand. The difference between plot a) and b) is hidden in the middle of the figure description in the end of a sentence. Would be better to have it in the description directly following a) resp. b), whereas "net forest gain at t=0" which is true for both plots should either be in the end or before the separation in a) and b). Or/additionally it could be mentioned as title in each plot whether it is primary to secondary or secondary to secondary. The axis title is only in plot a) but not in b) whereas the legend can be found in both plots. Please add the axis title to b) or remove

the legend from a) or do both and set the legend a bit aside, which would also help the reader to not confuse it with a second y-axis title at a first glance.

Please extend "Exponential carbon loss curve from (Hansis et al., 2015) and logarithmic gain curve from (Poorter et al., 2016) are used in this example" to something like "Exponential curve from Hansis et al., (2015) for carbon loss in all pools and gain in soil pool and logarithmic curve from Poorter et al., (2016) for gain in biomass pool are used in this example, which corresponds to the combinations C1 and C2 from Table 2 for a) and b) respectively."

---

## Referee Comment (RC2) · Anonymous Referee #2 · 28 Sep 2017

I read this manuscript with much interest, and found it to have novel elements which provide new and useful information. However, it could benefit from some revisions.

**General comments**

The first part of the paper is about land use changes, from forest to agriculture. However changes from primary and secondary forest to plantations are also discussed (and the abbreviation LUC is used). Harvesting in rotation is not generally considered land use change (but a land cover change), so this abbreviation might be better changed to Land cover change (LCC) which would encompass both the forest gain scenario and land use change (loss of forest to agriculture). The paper uses land cover data (Hansen), which further confuses the reader, when land use is mainly used. The authors could check the consistency of these terms (land use and land cover) in the

paper.

One of the concerns in the paper is the methods, which could be expanded to clarify some points. The analysis of the Hansen data, for example is not included. For example, what forest cover threshold did you use in the analysis? Did you for example mask out those pixels with loss or gain but with

mean, and do you have a reference for this?

You refer to "idealized scenarios". I am not sure about the choice of term here. Ideal for what?

Page 1 line 28. "compared against" could be changed to "compared to". Landsat is more commonly referred to as medium resolution (rather than high resolution), although the global maps are termed high resolution global maps. I would remove the term or would specify the resolution in m.

Page 2 line 7. Why "so-called"?

Page 8 line 5. could be rephrased: "are lower 20 years after the initial LUC" or in another way.

Page 7 Line 24. Is it necessary to describe a map as 'spatial'?

Page 7 line 14. Instead of "real world", "in a case study" or similar?

BGD

---

## Author Response (AR1)

**Response to comments**

**Paper #:** *bg-2017-291*

**Title:** Gross changes in forest area shape the future carbon balance of tropical forests **Journal:** Biogeosciences

**Reviewer #1:**

**General Comments:**

**Comment #1**

This is an interesting study, pointing out the importance of using gross instead of net land use transitions, distinguish between clearing of primary vs. secondary forest and to define a specific and reasonable time horizon when making land-based mitigation policies. Three main steps were taken: 1) the comparison between different response curves, 2) calculating different theoretical scenarios with a bookkeeping model to show the importance of considering gross forest area change and finding critical and 3) applying the ratio to real net land cover transitions from satellite data. Thereby, step 2 clearly takes the highest priority and consideration.

**Response #1**

We thank the reviewer for the comments and suggestions. Please see the detailed point-by-point responses below.

**Comment #2**

Still, some revision is needed: the abstract is very long which makes it partly difficult to get the main message of the study. Also, the gap in current research is not carved out very well (page 2, ln 16 ff, says that other models have already implemented gross transitions) and the objectives should become clearer. In the introduction a two-fold purpose of the study is mentioned, what about the 3rd step? What was its objective? The 3rd point cannot be found in the method section, it is just roughly described in the results. Thereby some steps remain unclear: e.g. the model considers LC transition to take place at time = t0, but the satellite covers a time series of 12 years. Are all the transitions during these years threatened as if they took place at one time t=0 and then the results for the different time horizons of 20, 50 and 100 years are calculated based on that? Or is the exact time of each transition considered and the time horizons starts to be calculated after the last transition took place? Or do the gross transitions in this case refer not to time (i.e. shift of one LC to another LC and back) but instead refer to transitions within the calculated gridcells of  $0.5^{\circ}$  resolution, as the satellite data was mentioned to have a 30 m resolution?

**Response #2**

We will shorten the abstract in the revised manuscript (reproduced in Response #4).

We will add sentences on **P2L21** to make the research gap and objectives more clear: "However, uncertainties in the simulated  $E_{LULCC}$  by grid-based DGVMs arise from the translation of the original LULCC datasets into plant functional type (PFT) maps and different processes comprised in different models (Arneth et al., 2017; Li et al., 2017). Although DGVMs are spatially and temporally explicit and include detailed physiological processes, the simulations using these models are time consuming and require long spin-up simulations, small time step calculations of biophysical effects and carbon fluxes, including processes less relevant to  $E_{LULCC}$ . Thus, DGVMs are not appropriate to perform, for instance, sensitivity tests for the assessment of LULCC carbon fluxes."

As suggested in **Comment #3**, we will include the  $3^{rd}$  step in the objectives in Introduction and also mention it in the Methods.

We will add sentences in the Methods to explain how we processed the 30 m forest cover data from Hansen et al. (2013): "Forest cover data from Hansen et al. (2013) comprise three layers at 30 m

resolution: tree cover fraction (0-100% in each pixel) in year 2000, forest area loss (each pixel labeled with a loss year) during 2000-2012, and forest gain during 2000-2012 (not specifying the gain year). Attributing the forest gain to a specific year is challenging because of the difficulty in detecting young forests from satellite reflectance measurements (Hansen et al., 2013). In this study, we used the forest loss and forest gain layers to calculate the ratios of gross-to-net area changes ( $\gamma_{Anet}^{Agross}$ ) at a 0.5° × 0.5° resolution, and  $\gamma_{Anet}^{Agross}$  represents the average values during 2000-2012 rather than for a single year since the year of forest gain is not reported. The gross changes at the 0.5° level were calculated by summing the absolute areas of forest loss and gain at the 30 m level during 2000-2012 in each 0.5° grid cell, while the net changes were the sum of gross forest loss (negative) and gross forest gain (positive)."

**Comment #3**

One possibility to handle the dominance of point 2 would be to make it to the only objective, and shift point 1 to the method section – the comparison seems to be anyhow just a plot of the different curves that justifies the usage of response curves based on Poorter instead of those from Houghton and Hansis. Another possibility would be to include 3 in the method section and give point 1 and 3 more weight - e.g. by calculating the critical gross to net ratios based on the Houghton and Hansis functions, applying it to the same grid cells and comparing it then with the results based on the Poorter function. This would be interesting outcome and extend the first objective of comparing the different response curves to more than just a simple plot of the different curves in the same graph. Further, it would be very interesting to not only know whether each gridcell was a sink or source but also to quantify the ELUC and sum it to total number – one if everything was primary forest at the first transition, and one as if all was secondary, and the same if the other response curves were used.

**Response #3**

We will include the 3rd step in the method section as suggested.

We calculated the critical ratios based on the exponential response curves from Hansis et al. (2015) (see **Figure R1** below) and compared the number of grid cells above the critical ratios with that using curves from Poorter et al. (2016) (see **Table R1** below). We didn't show the results from Houghton (1999) because the parameters of the exponential functions from Hansis et al. (2015) were already calibrated from the linear function of Houghton (1999).

As show in **Figure 1**, the equilibrium of secondary forest vegetation density with the recovery curve of Hansis et al. (2015) is higher than with Poorter et al. (2016) and we assumed that the same density of primary forest for both, and thus  $L_{\text{Hansis,primary}}(t) = L_{\text{Poorter,primary}}(t)$ ,  $L_{\text{Hansis,secondary}}(t) > L_{\text{Poorter,secondary}}(t)$ and  $G_{\text{Hansis}}(\infty) > G_{\text{Poorter}}(\infty)$ . Note that a positive value of carbon flux indicates carbon emission to the atmosphere. Combined with **Eq (7)** in the manuscript,  $\gamma_{Anet}^{Agross} = \frac{L(t) - G(t)}{L(t) + G(t)}$ , the different equilibrium states of secondary forest vegetation can explain the differences of critical ratios over time between Hansis et al. (2015) and Poorter et al. (2016) in **Figure R1**. Accordingly, a higher critical ratio leads to smaller number of  $0.5^{\circ} \times 0.5^{\circ}$  grid cells with  $\gamma_{Anet}^{Agross}$  beyond the critical ratio (**Table R1**). **Figure R1** The critical value of  $\gamma_{Anet}^{Agross}$  at which  $\Sigma E_{LULCC,gross}$  is zero, going from a net source to a net sink with different time horizons, using the biomass recovery response curves from Poorter et al. (2016) (solid, same as **Figure 4c**) and from Hansis et al. (2015) (dashed). Values larger than this critical value indicate that the initial forest area change has the net cumulative effect to emit CO2 at a given time-horizon on the x-axis. Note the different y-axis scale. The lower critical ratio values in the case of primary forest initial loss is because primary forests have a larger biomass, so that a small gross-to-net initial change in forest area will legate a source at a given horizon than if secondary forests are initially lost.

**Table R1** Number of  $0.5^{\circ} \times 0.5^{\circ}$  grid cells with  $\gamma_{Anet}^{Agross}$  above the critical ratio for which the system is a net cumulative source of CO2 to the atmosphere, for different time horizons. The calculation was done using the biomass recovery response curves from Hansis et al. (2015) and Poorter et al. (2016) in Latin America. The values of  $\gamma_{Anet}^{Agross}$  were calculated based on high-resolution net and gross forest area change data from Hansen et al. (2013) during 2000-2012. Secondary-to-secondary represents a net forest gain with gross secondary forest loss by assuming that all lost forests were secondary, and primary-to-secondary represents a net forest gain with gross by assuming that all lost forest loss by assuming that all lost forests were primary.

|               |                | 20 yr          |                     | 50 yr          |                     | 100 yr         |                     |
|---------------|----------------|----------------|---------------------|----------------|---------------------|----------------|---------------------|
|               |                | Critical ratio | Grid cell
number | Critical ratio | Grid cell
number | Critical ratio | Grid cell
number |
| secondary-to- | Poorter et al. | 7.2            | 102                 | 22.5           | 42                  | -              | -                   |
| secondary     | Hansis et al.  | 4.2            | 143                 | 15.3           | 57                  | 97.4           | 9                   |
| primary-to-   | Poorter et al. | 2.4            | 199                 | 3.1            | 175                 | 3.7            | 155                 |
| secondary     | Hansis et al.  | 2.5            | 198                 | 4.1            | 147                 | 5.0            | 126                 |

As suggested, we also calculated the total amount of  $\Sigma E_{LULCC}$  of the grid cells beyond the critical ratios with different time horizons in **Figure 5**. The numbers are given in the **Table R2** below and we will incorporate it in the revised manuscript.

Taking C1 secondary-to-secondary at 20 yr horizon for example, using net transitions results in a carbon sink of 12 Tg C but using gross transitions results in a carbon emission of 21 Tg C in the grid cells with  $\gamma_{\text{Anet}}^{\text{Agross}} > 7.2$  (Figure 5).

**Table R2** Cumulative carbon flux (Tg C) using gross transitions ( $\Sigma E_{LULCC,gross}$ ) and net transitions ( $\Sigma E_{LULCC,net}$ ) in the grid cells with  $\gamma_{Anet}^{Agross}$  beyond the critical ratios at different time horizons. The gross

and net forest area changes are based on the data from Hansen et al. (2013). Positive value of carbon flux indicates carbon emission to the atmosphere. Secondary-to-secondary represents a net forest gain with gross secondary forest loss (C1) by assuming that all lost forests were secondary, and primary-to-secondary represents a net forest gain with gross primary forest loss (C2) by assuming that all lost forests were primary.

| Tg C         | C1: secondary-to-secondary |                          |                        | C2: pri        | mary-to-secon            | ndary                  |
|--------------|----------------------------|--------------------------|------------------------|----------------|--------------------------|------------------------|
| Time horizon | Critical ratio             | $\Sigma E_{LULCC,gross}$ | $\Sigma E_{LULCC,net}$ | Critical ratio | $\Sigma E_{LULCC,gross}$ | $\Sigma E_{LULCC,net}$ |
| 20 yr        | 7.2                        | 21                       | -12                    | 2.4            | 162                      | -38                    |
| 50 yr        | 22.5                       | 3                        | -2                     | 3.1            | 125                      | -39                    |
| 100 yr       | -                          | -                        | -                      | 3.7            | 99                       | -36                    |

We will add these new analyses suggested by the reviewer in the revised manuscript.

**Specific Comments:**

**Comment #4**

Abstract: The abstract should be shortened to better focus on the findings, which would make it easier to read and understand. E.g. is the 3rd sentence really relevant for the findings of this study? Especially also from line 19 to 27 there might be possibilities to shorten, summarize and simplify. Where shapes of the three different curves relevant for the finding? The finding here is difficult to understand, the sentences a bit complicated and several sentences basically say the same: You found and show critical values of gross to net forest area change above which ELUC of a net a net forest area gain switches from  $CO_2$  sink to source.

**Response #4**

As suggested, we will shorten the Abstract (160 words less) as follows: "

Bookkeeping models are used to estimate land-use and land-cover change (LULCC) carbon fluxes  $(E_{LULCC})$ . The uncertainty of bookkeeping models partly arises from data used to define response curves (usually from local data) and their representativeness for application to large regions. Here, we compare biomass recovery curves derived from a recent synthesis of secondary forest plots in Latin America by Poorter et al. (2016) with the curves used previously in bookkeeping models from Houghton (1999) and Hansis et al. (2015). We find that the two latter models overestimate the longterm (100 years) biomass carbon density of secondary forest by about 25%. We also use idealized LULCC scenarios combined with these three different response curves to demonstrate the importance of considering gross forest area changes instead of net forest area changes for estimating regional ELULCC. In the illustrative case of a net gain in forest area composed of a large gross loss and a large gross gain occurring during a single year, the initial gross loss has an important legacy effect on  $E_{LULCC}$  so that the system can be a net source of  $CO_2$  to the atmosphere long after the initial forest area change. We show the existence of critical values of the ratio of gross area change over net area change ( $\gamma_{Anet}^{Agross}$ ) above which cumulative  $E_{LULCC}$  is a net CO2 source rather than a sink for a given time horizon after the initial perturbation. These theoretical critical ratio values derived from simulations of a bookkeeping model are compared with real-world observations from the 30 m resolution Landsat TM data of gross and net forest area change in the Amazon. This allows us to diagnose areas where current forest gains with a large land turnover will still legate LULCC carbon emissions in 20, 50 and 100 years.

**"**

**Comment #5**

Introduction: Page 2, ln 8: that is for DGVMs the sub-grid transitions that sum up to net changes, here a reference to e.g. Bayer et al. 2017: doi:10.5194/esd-8-91-2017 could be nice, who focused on the problematic of sub-grid transitions.

**Response #5**

We will add this reference in the revised manuscript.

**Comment #6**

3.4 Page 7, ln 14: "we pose the question whether such ratios can be observed in the real world" – but this is not what you are answering with your approach. As far as I understood you just calculate using your rates, whether the regions are a sink or a source.

**Response #6**

This sentence on **P7L14** will be revised as: "...we further combined such ratios with the land use and land cover change datasets to determine whether a region is a carbon sink or source at a given time horizon."

**Comment #7**

Page 9, ln 22: "With a too high rotation rate of forests, i.e. a large gross to net area change ratio, a net forest gain could still legate a net carbon source over a long period in the future." I don't agree, as I think long rotation secondary forests should have other response curves than short rotation forest, as short rotation forest don't store as much carbon that can be lost afterwards.

**Response #7**

This sentence on **P9L22** will be revised as: "With a large gross to net area change ratio, a net forest gain could still legate a net carbon source over a long period in the future."

**Comment #8**

4. Discussion: You state that the response curves used in bookkeeping models from Houghton (1999) and Hansis et al. (2015) overestimate carbon density – that implies that Porters values are true, while Houghton and Hansis are wrong. But also Houghton and Hansis are based on measurements, right? Maybe just not in the right region? It would be helpful to mention in the discussion where the measurements for Houghton and Hansis models were located.

**Response #8**

We showed the differences in biomass recovery curves between Poorter et al. (2016) and Houghton (1999) and Hansis et al. (2015), but we didn't say that it implies "Porters values are true, while Houghton and Hansis are wrong." We only argued that it may bias in this particular region where Poorter et al.'s field survey covers. The reasons for these differences could be the assumptions made for secondary forest by Houghton et al. (1983), the number of field sites and the different locations where field measurements were conducted, as the reviewer said.

We will add some sentences on **P8L9** to clarify it: "The biomass recovery curves of Neotropical secondary forests from Poorter et al. (2016) are lower 20 years after the initial perturbation than those used in the bookkeeping models of Houghton (1999) and Hansis et al. (2015), implying that these models simulate different LULCC carbon fluxes in Latin America from those using the recovery curves of Poorter et al. (2016). The carbon density in undisturbed forests in the bookkeeping models of Houghton (1999) and Hansis et al. (2015) were essentially based on Whittaker and Likens (1973), multiplied by a factor of 0.75 to approximate the lower carbon density of secondary forests (Houghton et al., 1983). The carbon density data from Whittaker and Likens (1973) are subject to two sources of uncertainty. First, these values represent biomass in the 1950s (Woodwell et al., 1978) rather than present days, and second, they were compiled from very limited field measurements for tropical forests. In fact, Whittaker and Likens (1973) claimed in their study that data "for tropical communities are very meager" and the mean biomass density is "a subjectively chosen intermediate value based on very few measurements" to avoid extreme values."

**Reference**

Houghton, R. A., Hobbie, J. E., Melillo, J. M., Moore, B., Peterson, B. J., Shaver, G. R. and Woodwell, G. M.: Changes in the Carbon Content of Terrestrial Biota and Soils between 1860 and 1980: A Net Release of CO" 2 to the Atmosphere, Ecol. Monogr., 53(3), 235–262, doi:10.2307/1942531, 1983.

Whittaker, R. H. and Likens, G. E.: Carbon in the biota., in Brookhaven symposia in biology, pp. 281–302., 1973.

Woodwell, G. M., Whittaker, R. H., Reiners, W. a, Likens, G. E., Delwiche, C. C. and Botkin, D. B.: The biota and the world carbon budget, Science, 199(4325), 141–146, doi:10.1126/science.199.4325.141, 1978.

**Comment #9**

Is the extent to which gross versus net transitions affect ELUC comparable what other studies investigating gross versus net transitions studied? You mention several studies about this issue which were performed with carbon models – they should also appear in the discussion, showing how your results compare with what they found.

**Response #9**

We will add some discussion about the impacts of gross and net transitions on  $E_{LULCC}$  on **P8L13**: "Some DGVMs (Shevliakova et al., 2009; Stocker et al., 2014; Wilkenskjeld et al., 2014; Yue et al., 2017; Bayer et al., 2017) as well as a bookkeeping model (Hansis et al., 2015) have implemented gross land use and land cover transitions, and thus simulated a higher  $E_{LULCC}$  than using net transitions. Arneth et al. (2017) reviewed the "missing processes" in LULCC modeling by DGVMs and found that ignoring gross LULCC could underestimate the global cumulative  $E_{LULCC}$  by 36 Pg C on average over the historical period (1901-2014).". However, the  $E_{LULCC}$  from land carbon models cannot be directly compared with  $E_{LULCC}$  from bookkeeping models, because of the different processes in models and the different definitions of  $E_{LULCC}$ . In addition, our study focused on the ratios of gross-to-net changes rather than the estimates of  $E_{LULCC}$ , and thus it is difficult and not necessary to compare with  $E_{LULCC}$  from land carbon models.

**Comment #10**

**Technical corrections**

General: In all figures and the text, it might be useful to replace the term "biomass carbon density" by "vegetation carbon density", as biomass is less well defined and includes in some disciplines also dead biomass and soil microbial biomass which would count here to the soil pool.

**Response #10**

We will revise it accordingly.

**Comment #11**

Page 1, ln 26: "critical value" should be plural

**Response #11**

We will revise it accordingly.

**Comment #12**

Page 2, Ln 10: "Gross LUC occurs in tropical regions with shifting cultivation (Hurtt et al., 2011) but also everywhere forests are cut and new plantations created at the same time" Is here a "where" after "everywhere" missing?

**Response #12**

This sentence on **P2L10** will be revised as: "Gross LULCC occurs in tropical regions with shifting cultivation (Hurtt et al., 2011) and also in other regions where forests are cut and new plantations created at the same time."

**Comment #13**

Table 1: The table caption uses gamma gross to net but in the table heading and the text gamma Agross to Anet is used.

**Response #13**

We will revise it accordingly.

**Comment #14**

Fig 1: biomass from primary forest: reference missing; Legend for a) and b) shows biomass, which can only be found in plot c) that has an own legend.

**Response #14**

We will add the reference in the legend and remove biomass from the legend in (a) and (b).

Fig 3: include dashed and solid line in legend. In figure description logarithmic asymptotic should be removed or referred to both - solid and dashed, so it becomes more clear that there is no difference in the response curve between solid and dashed, but just which systems are transformed.

**Response #15**

We will add it in the legend and revise the caption.

**Comment #15**

Fig 4 is a bit difficult to understand. The difference between plot a) and b) is hidden in the middle of the figure description in the end of a sentence. Would be better to have it in the description directly following a) resp. b), whereas "net forest gain at t=0" which is true for both plots should either be in the end or before the separation in a) and b). Or/additionally it could be mentioned as title in each plot whether it is primary to secondary or secondary to secondary. The axis title is only in plot a) but not in b) whereas the legend can be found in both plots. Please add the axis title to b) or remove the legend from a) or do both and set the legend a bit aside, which would also help the reader to not confuse it with a second y-axis title at a first glance.

Please extend "Exponential carbon loss curve from (Hansis et al., 2015) and logarithmic gain curve from (Poorter et al., 2016) are used in this example" to something like "Exponential curve from Hansis et al., (2015) for carbon loss in all pools and gain in soil pool and logarithmic curve from Poorter et al., (2016) for gain in biomass pool are used in this example, which corresponds to the combinations C1 and C2 from Table 2 for a) and b) respectively."

**Response #16**

As suggested, we will revise the caption and re-plot the figure (reproduced below).

Figure 4 Time evolution of cumulative carbon flux ( $\Sigma E_{LULCC,gross}$ ) after an initial forest area change involving gross forest area changes followed by no forest area change. The three panels show results of our bookkeeping model for three case studies (a) a net forest gain at t = 0 with initial secondary forest loss followed by secondary forest regrowth (secondary-to-secondary, C1 in Table 2), (b) the same net area gain at t = 0 with initial primary forest loss followed by secondary forest regrowth (primary-to-secondary C2 in Table 2), and (c) the critical value of  $\gamma_{Anet}^{Agross}$  at which  $\Sigma E_{LULCC,gross}$  is zero, going from a net source to a net sink for different time horizon in the x-axis. The colored curves in (a) and (b) have the same net area change  $(A_{net} = +1 ha)$  at t = 0 but variable values of the initial gross-tonet area change ratios ( $\gamma_{Anet}^{Agross}$ ). The red line in (a) and (b) is the zero line, defining the time after initial forest area change at which the system reaches a neutral carbon balance. The light and dark green lines in (c) represent the critical ratios for a net initial forest gain scenario with secondary-tosecondary (a) and primary-to-secondary (b) gross forest area change, respectively. Values larger than this critical value indicate that the initial forest area change has the net effect to emit  $CO_2$  for a given time horizon in the x-axis. Exponential curve from Hansis et al. (2015) for carbon loss in all pools and gain in soil pool and logarithmic curve from Poorter et al. (2016) for gain in biomass pool are used in this example.

**Reviewer #2:**

**General Comments:**

**Comment #1**

I read this manuscript with much interest, and found it to have novel elements which provide new and useful information. However, it could benefit from some revisions.

**Response #1**

We thank the reviewer for the comments and suggestions. Please see the detailed point-by-point responses below.

**Comment #2**

The first part of the paper is about land use changes, from forest to agriculture. However changes from primary and secondary forest to plantations are also discussed (and the abbreviation LUC is used). Harvesting in rotation is not generally considered land use change (but a land cover change), so this abbreviation might be better changed to Land cover change (LCC) which would encompass both the forest gain scenario and land use change (loss of forest to agriculture). The paper uses land cover data (Hansen), which further confuses the reader, when land use is mainly used. The authors could check the consistency of these terms (land use and land cover) in the paper.

**Response #2**

We agree that the satellite data from Hansen et al. (2013) we used in the case study is land cover change rather than land use change, and the idealized scenarios are more land use change although also a land cover change, as described on **P3L12** "The land-use changes considered in this study are forest loss (tropical moist forest transformed to cropland) and forest gain (cropland abandonment to secondary tropical moist forest) in Latin America.". We will change the term into "land-use and land-cover change (LULCC)" throughout in the text to be consistent.

**Comment #3**

One of the concerns in the paper is the methods, which could be expanded to clarify some points. The analysis of the Hansen data, for example is not included. For example, what forest cover threshold did you use in the analysis? Did you for example mask out those pixels with loss or gain but with <10%, or another appropriate canopy cover threshold for the region? Or is it exactly following the Poorters map? How was the change of grid cell to 0.50 done? For example, pixels only partially within the area of interest are included or not? I wonder if the choice of grid cell size would impact the results? Was 0.50 chosen for a specific reason?

**Response #3**

We will add some sentences to clarify the forest cover change data from Hansen et al. (2013) in the revised manuscript: "Forest cover data from Hansen et al. (2013) comprise three layers at 30 m resolution: tree cover fraction (0-100% in each pixel) in year 2000, forest area loss (each pixel labeled with a loss year) during 2000-2012, and forest gain during 2000-2012 (not specifying the gain year). Attributing the forest gain to a specific year is challenging because of the difficulty in detecting young forests from satellite reflectance measurements (Hansen et al., 2013). In this study, we used the forest loss and forest gain layers to calculate the ratios of gross-to-net area changes ( $\gamma_{Aret}^{Agross}$ ) at a 0.5° × 0.5° resolution, and  $\gamma_{Aret}^{Agross}$  represents the average values during 2000-2012 rather than for a single year since the year of forest gain is not reported. The gross changes at the 0.5° level were calculated by summing the absolute areas of forest loss and gain at the 30 m level during 2000-2012 in each 0.5° grid cell, while the net changes were the sum of gross forest loss (negative) and gross forest gain (positive)." Thus, we didn't use the tree cover fraction threshold because we didn't use the tree fraction data.

It is not necessary to be exactly the same region of the Poorter et al.'s map because the biomass recovery estimates from Poorter et al. (2016) are based on forest sites and forest plots and thus represent a rough (not precise) Latin America region. Thus there is no such issue of partially

overlapped pixels. We gave the latitudes and longitudes of the region we used from the map of Hansen et al. (2013) in **Figure 5**.

The gross changes compared to net changes essentially is a matter of resolution. For example, if the source data is at 30 m spatial resolution and all the models are run at 30 m resolution, there would be no difference between gross and net changes. The differences between gross and net changes only emerge when aggregating high-resolution data into a coarser resolution. The reason for choosing the 0.5° resolution was described on **P7L18**: "The spatial resolution of  $0.5^{\circ}$  is a typical resolution of **DGVMs** when they simulate global  $E_{LULCC}$ ." Because the 30 m spatial resolution from Hansen et al.'s data is relatively high, using other grid cell size like 0.1° or 1° would be expected to give similar patterns as using 0.5° in **Figure 5**.

**Comment #4**

Figure 5 is also not clear to me, for example (if I understand correctly), those pixels in blue reached the threshold for the secondary forest clearing (and also the primary forest clearing) and those in green reached the threshold for the primary forest clearing only? This would be useful information to include in the caption.

**Response #4**

Yes, that is correct. We will add it in the caption as suggested: "The blue grid cells represent a cumulative carbon emission in 20 years no matter whether the lost forest is primary or secondary. The green ones represent a cumulative carbon emission only if the cleared forests are primary forests."

**Comment #5**

The results for the soil carbon change are also interesting and useful to include, but I find the discussion about this lacking. Indeed, there is a huge amount of uncertainty related to changes in soil carbon (see for example Don et al. 2011 Impact of tropical land-use change on soil organic carbon stocks – a meta-analysis). Incorporating some aspect of uncertainties related to this could have been helpful, and indeed, uncertainties are missing in all findings of the paper.

**Response #5**

We will revise the sentences on **P8L9**: "Differences may also exist for soil carbon dynamics after LULCC. There are a great number of meta-analyses or reviews (e.g. Davidson & Ackerman, 1993; Post & Kwon, 2000; Conant et al., 2001; Paul et al., 2002; Davis & Condron, 2002; Guo & Gifford, 2002; Murty et al., 2002; West et al., 2004; Laganière et al., 2010; Poeplau et al., 2011; Powers et al., 2011; Don et al., 2011; Li et al., 2012; Marín-Spiotta & Sharma, 2013; Wei et al., 2014; Kurganova et al., 2014) on the soil carbon change after LULCC based on field measurement data (mostly paired sites and chronosequences). These studies may generally agree with the directions of soil carbon change after LULCC (e.g. soil carbon loss after forest clearing for cropland), but the magnitudes and temporal dynamics of soil carbon changes remain highly uncertain because, among other things, of the limited site number and the diversity of soil properties. Field measurements at site level may be unrepresentative of the whole region because the distribution of biophysical conditions like soil texture, precipitation and temperature may not match the distribution of the whole set of such factors in the LULCC areas in a given region (Powers et al., 2011)."

**Specific Comments:**

**Comment #6**

Page 7, line 14-16. There are a number of datasets which you could use, and the data also do not limit the work to small scale analysis, so this sentence seems not to be useful.

**Response #6**

This sentence will be deleted.

**Comment #7**

Page 3 line 29/30. I would include here or somewhere appropriate, some numbers related to the total biomass used in the paper from Poorter.

**Response #7**

The number related to the ratio of aboveground to total biomass is only mentioned in the supporting information in Poorter et al. (2016). The ratio Poorter et al. (2016) used is from FAO FRA, which is 0.82, basically the same as we used (0.81). We will revise the sentence about on **P3L30**: "For both response curves, a ratio of 0.81 (Liu et al., 2015; Peacock et al., 2007; Saatchi et al., 2011) was used to convert aboveground biomass reported by Poorter et al. (2016) to total biomass, and this ratio is consistent with the one (0.82) that Poorter et al. (2016) used based on FAO FRA (2010)."

**Comment #8**

Page 9 line 9 – the "new planted forest in rotation practice"- it is not clear what you mean, and do you have a reference for this?

**Response #8**

We will revise this sentence as: "Forest management practices like wood harvest and thinning extract carbon from the ecosystem and release it to the atmosphere (Houghton et al., 2012), while recovering secondary forest from past deforestation and logging (Pan et al., 2011) and even old-growth forests (Luyssaert et al., 2008) can act as carbon sinks."

**Reference**

Luyssaert, S., Schulze, E.-D., Börner, A., Knohl, A., Hessenmöller, D., Law, B. E., Ciais, P. and Grace, J.: Old-growth forests as global carbon sinks, Nature, 455(7210), 213–215, doi:10.1038/nature07276, 2008.

Pan, Y., Birdsey, R. A., Fang, J., Houghton, R., Kauppi, P. E., Kurz, W. A., Phillips, O. L., Shvidenko, A., Lewis, S. L., Canadell, J. G., Ciais, P., Jackson, R. B., Pacala, S. W., McGuire, A. D., Piao, S., Rautiainen, A., Sitch, S. and Hayes, D.: A large and persistent carbon sink in the world's forests., Science, 333(6045), 988–993, doi:10.1126/science.1201609, 2011.

**Comment #9**

You refer to "idealized scenarios". I am not sure about the choice of term here. Ideal for what?

**Response #9**

We think this is a matter of English here. "idealized" refers to "conceptual" while "ideal" is more like "optimal". Because we want to demonstrate the difference between gross and net changes on  $E_{LULCC}$  and determine the critical gross-to-net change ratio, we used these idealized scenarios that are simple and representative, and may not the case in reality.

Page 1 line 28. "compared against" could be changed to "compared to". Landsat is more commonly referred to as medium resolution (rather than high resolution), although the global maps are termed high resolution global maps. I would remove the term or would specify the resolution in m.

**Response #10**

We will revise it accordingly.

**Comment #10**

Page 2 line 7. Why "so-called"?

**Response #11**

We will delete it accordingly.

**Comment #11**

Page 8 line 5. could be rephrased: "are lower 20 years after the initial LULCC" or in another way.

**Response #12**

We will revise it accordingly.

**Comment #12**

Page 7 Line 24. Is it necessary to describe a map as 'spatial'?

**Response #13**

We will delete it accordingly.

**Comment #13**

Page 7 line 14. Instead of "real world", "in a case study" or similar?

**Response #14**

This sentence on **P7L14** will be revised as: "...we further combined such ratios with the land use and land cover change datasets to determine whether a region is a carbon sink or source at a given time horizon."

**Gross changes in forest area shape the future carbon balance of tropical forests**

Wei Li1, Philippe Ciais1, Chao Yue1, Thomas Gasser2, Shushi Peng3, Ana Bastos1

1Laboratoire des Sciences du Climat et de l'Environnement, LSCE/IPSL, CEA-CNRS-UVSQ, Université Paris-Saclay, 91191
 Gif-sur-Yvette, France
 2International Institute for Applied Systems Analysis (IIASA), A-2361 Laxenburg, Austria
 3Sino-French Institute for Earth System Science, College of Urban and Environmental Sciences, Peking University,

Sino-French Institute for Earth System Science, College of Urban and Environmental Sciences, Peking University, Beijing 100871, China

Correspondence to: Wei Li (wei.li@lsce.ipsl.fr)

- 10 Abstract. Bookkeeping models are used to estimate land-use and land-cover change (LULCC) carbon fluxes ( $E_{LULCC}$ ). The uncertainty of bookkeeping models partly arises from data used to define response curves (usually from local data) and their representativeness for application to large regions. Here, we compare biomass recovery curves derived from a recent synthesis of secondary forest plots in Latin America by Poorter et al. (2016) with the curves used previously in bookkeeping models from Houghton (1999) and Hansis et al. (2015). We find that the two latter models overestimate the long-term (100 years)
- 15 vegetation carbon density of secondary forest by about 25%. We also use idealized LULCC scenarios combined with these three different response curves to demonstrate the importance of considering gross forest area changes instead of net forest area changes for estimating regional  $E_{LULCC}$ . In the illustrative case of a net gain in forest area composed of a large gross loss and a large gross gain occurring during a single year, the initial gross loss has an important legacy effect on  $E_{LULCC}$  so that the system can be a net source of CO2 to the atmosphere long after the initial forest area change. We show the existence of critical
- 20 values of the ratio of gross area change over net area change ( $\gamma_{\text{Anset}}^{\text{Anset}}$ ), above which cumulative  $E_{\text{LULCC}}$  is a net CO2 source rather than a sink for a given time horizon after the initial perturbation. These theoretical critical ratio values derived from simulations of a bookkeeping model are compared with real-world observations from the 30 m resolution Landsat TM data of gross and net forest area change in the Amazon. This allows us to diagnose areas where current forest gains with a large land turnover will still legate LULCC carbon emissions in 20, 50 and 100 years.

1

25

Deleted: Bookkeeping models are used to estimate land-use change (LUC) carbon fluxes ( $E_{LUC}$ ). These models combine time series of areas subject to different LUC types with response curves of carbon pools in ecosystems and harvested products after a unit change of land use. The level of detail of bookkeeping models depends on the number of response curves used for different regions the carbon pools they represent, and the diversity of LUC types considered. The uncertainty of bookkeeping models arises from data used to define response curves (usually local data) and their representativeness of large regions. Here, we compare biomass recovery curves derived from a recent synthesis of secondary forest plots data by Poorter et al. (2016) with the curves used in bookkeeping models from Houghton (1999) and Hansis et al. (2015) in Latin America. We find that both Houghton (1999) and Hansis et al. (2015) overestimate the long-term (100 years) biomass carbon density of secondary forest, by about 25%. We also show the importance of considering gross forest area change in addition to the net forest area change for estimating regional  $E_{LUC}$ . To do so, simulations are constructed with a bookkeeping model calibrated with three different sets of response curves (linear, exponential and logarithmic) to study  $E_{\rm LUC}$  created by a pulse of net forest area change, with different gross-to-net forest area change ratios (γAgross Anet). Following the initial pulse of forest area change, Eurc is subsequently calculated over 100 years. Considering a region subject to a net gain in forest area during one year, different values of gross forest area changes that sum up to this initial net gain can change the magnitude and even the sign of  $E_{\rm LUC}$  with a given time horizon after the initial forest area change. In other words, in the case of a net gain in forest area composed of a large gross loss and a large gross gain, the initial gross loss has an important legacy effect that the system can be a net source of CO2 to the atmosphere. We show the existence of a critical value of  $\gamma$ Agross Anet above which ELUC switches from CO2 sink to source with a given time horizon after the initial forest area change. This critical ratio derived from the structure of the bookkeeping model is compared against realworld high resolution Landsat TM observations of gross forest area change in the Amazon to distinguish areas where current forest land turnover will legate LUC carbon emissions or sinks in 20 years, 50 years and 100 years in the future.

**1** Introduction**

The global carbon flux from land-use and land-cover change ( $E_{LULCC}$ ) represents a net source of carbon to the atmosphere of 0.9 ± 0.5 Gt C yr-1 during the last decade (Ciais et al., 2013; Le Quéré et al., 2015).  $E_{\underline{LULCC}}$  is usually estimated by bookkeeping models (Hansis et al., 2015; Houghton, 2003), dynamic global vegetation models (DGVMs) (Le Quéré et al., 2015; Sitch et

- 5 al., 2015) or compact earth system models (Gasser et al., 2017). Most DGVMs (e.g. in the TRENDY project, Sitch et al., 2015) estimate emissions due only to net area changes between different land-use / land-cover types in a grid cell. At the moment, efforts are being made to incorporate gross land-use and land-cover change (LULCC) in these models, that is for DGVMs the sub-grid transitions that sum up to net changes (Bayer et al., 2017). The bookkeeping model of Houghton (1999) includes emissions from both net area changes and gross LULCC from shifting cultivation, previously at the scale of large regions
- 10 (Houghton, 2003), and more recently for each country (Houghton and Nassikas, 2017). Gross LULCC occurs in tropical regions with shifting cultivation (Hurtt et al., 2011) and also in other regions where forests are cut and new plantations created at the same time. For example, consider a region with co-existing forest and crops where 20% of the land is converted from primary forest to crops while 20% sees crop abandonment to forest in the same period. The net change corresponds to a stable forest area, but the large carbon loss from primary forest is not compensated by the small carbon gain of the new plantations.
- 15 In this example, the region will be a net source of  $CO_2$  during several years. Because of the non-symmetrical dynamics of  $CO_2$ fluxes between forest loss and gain,  $E_{\underline{LULCC}}$  differs between net and gross area changes. Arneth et al. (2017) recently reviewed this issue using DGVMs and concluded that considering gross  $\underline{LULCC}$  significantly increased the simulated  $E_{\underline{LULCC}}$  at global scale. Gross land-use area transition datasets including e.g. shifting cultivation practice (Hurtt et al., 2011) and reconstructions using empirical ratios between gross and net transitions (Fuchs et al., 2015) are now available and have been implemented in
- 20 a bookkeeping model (Hansis et al., 2015) as well as in some DGVMs to improve the estimate of ELULCC (Fuchs et al., 2016; Shevliakova et al., 2009; Stocker et al., 2014; Wilkenskjeld et al., 2014; Yue et al., 2017). However, uncertainties in the simulated ELULCC by grid-based DGVMs arise from the translation of the original LULCC datasets into plant functional type (PFT) maps and different processes comprised in different models (Arneth et al., 2017; Li et al., 2017). Although DGVMs are spatially and temporally explicit and include detailed physiological processes, the simulations using these models are time
- 25 consuming and require long spin-up simulations, small time step calculations of biophysical effects and carbon fluxes, including processes less relevant to E4.ULCC. Thus, DGVMs are not appropriate to perform, for instance, sensitivity tests for the assessment of LULCC carbon fluxes.

Bookkeeping models use response curves for biomass and soil carbon stocks consecutive to LULCC disturbance and timeseries of LULCC areas to estimate ELULCC (Hansis et al., 2015; Houghton, 1999). Response curves can be linear (Houghton,

30 1999, 2003), exponential (Hansis et al., 2015) or of other types. The carbon densities of different land-use types are derived from field measurements (Houghton et al., 1983). Even though carbon densities have a high spatial variability in the real world, the same response curve measured at one location is often applied in bookkeeping models over large regions. A recent study of the biomass resilience of secondary forests in the Neotropics provides new biomass recovery curves from 45 secondary

| Deleted                                             | LUC                                                                                                                  |
|-----------------------------------------------------|----------------------------------------------------------------------------------------------------------------------|
|                                                     |                                                                                                                      |
|                                                     |                                                                                                                      |
|                                                     |                                                                                                                      |
| Deleted                                             |                                                                                                                      |
| Deleted                                             | so-called                                                                                                            |
|                                                     |                                                                                                                      |
| Deleted                                             | LUC                                                                                                                  |
|                                                     |                                                                                                                      |
|                                                     |                                                                                                                      |
| Deleted                                             |                                                                                                                      |
| cultivation                                         | (Hurtt et al 2011) but also everywhere forests are c                                                                 |
| and new p                                           | lantations created at the same time.                                                                                 |
|                                                     |                                                                                                                      |
|                                                     |                                                                                                                      |
|                                                     |                                                                                                                      |
| Deleted                                             | LUC                                                                                                                  |
| Deleted                                             | LUC                                                                                                                  |
| Deleted
Deleted
Deleted                       | LUC
LUC                                                                                                           |
| Deleted
Deleted
Deleted                       | LUC
LUC                                                                                                           |
| Deleted
Deleted
Deleted                       | LUC
LUC                                                                                                           |
| Deleted
Deleted
Deleted                       | LUC
LUC
LUC
LUC                                                                                             |
| Deleted
Deleted
Deleted
Deleted
Deleted | LUC
LUC
LUC
Luc
Iand carbon models
Luc                                                                |
| Deleted
Deleted
Deleted
Deleted
Deleted | LUC
LUC
LUC
Luc
Land carbon models
Luc
Luc
Luc
Luc
Luc
Luc
Luc
Luc
Luc
Luc |
| Deleted
Deleted
Deleted
Deleted
Formatt | LUC
LUC
LUC
Luc
Land carbon models
Luc
Luc
Luc
Luc
Luc
Luc
Luc
Luc
Luc
Luc |
| Deleted
Deleted
Deleted
Deleted
Formatt | LUC
LUC
LUC
Luc
Luc
Luc
Luc
Luc
Luc
Luc
Luc
Luc
Luc                              |
| Deleted
Deleted
Deleted
Deleted
Formatt | LUC
LUC
LUC
: LUC
: Luc
: land carbon models
: LUC
ed: Subscript                                |

| Deleted: LUC |  |
|--------------|--|
| Deleted: LUC |  |
| Deleted. LOC |  |
| Deleted: LUC |  |
|              |  |

forest sites (Poorter et al., 2016). These new data are valuable to revisit the response curves for the regrowth of secondary forest in the Amazon area, an important region with a large ELULCC:

In this study, we first aim to compare the recent biomass regrowth curves from Poorter et al. (2016) with the ones used in two bookkeeping models (Hansis et al., 2015; Houghton, 1999) for their implications in Eq.ULCC. Second, we demonstrate that because of the asymmetry between carbon loss from deforestation and carbon gains from regrowth, even when the net forest 5 area change is positive, a large initial gross forest area change can still cause Eq.ULCC to be a source of CO2 to the atmosphere on multi-decadal horizons. Last, we apply our conceptual calculation to the satellite forest data to diagnose areas with net forest gains but cumulative LULCC carbon emissions.

Based on ELULCC calculated using a bookkeeping approach and several idealized scenarios constructed to have different gross 10 forest area changes but with the same net area change (Section 3.2), we show the existence of a critical ratio of gross-to-net forest area change above which cumulativ

---

## Author Response (AR2)

**Response to comments**

**Paper #:** *bg-2017-291*
**Title:** *Gross changes in forest area shape the future carbon balance of tropical forests*
**Journal:** *Biogeosciences*

**Editor:**

**General Comments:**

**Comment #1**

The authors conscientiously addressed referee comments and I thank them for that but I would appreciate if they gave the manuscript another careful read for accuracy noting as one example 'legate' instead of 'negate' in the abstract. I am also of the opinion that Figure 2 could be improved to avoid text overlaps in the figure and that the light green of Figure 3 would benefit from a more readable color. Please make these and any other changes that you see fit to improve the revision and I would be happy to consider it for publication in Biogeosciences.

**Response #1**

We thank the editor for handling our manuscript and the comments. As suggested, we proofread the manuscript carefully for accuracy. "legate" is a term referring the legacy land-use change (LUC) flux from the slow processes after LUC (e.g. biomass recovery and soil carbon decomposition). We are aware that this word is too technical for the broad audience, and thus rephrase the related sentences accordingly.

As suggested, we improved Figure 2 to avoid text overlaps.

There is no light green in Figure 3, but we assumed the editor referring to Figure 4. We thus changed the light green color in Figure 4.

[revised manuscript text omitted]